# The curled wake model: A three-dimensional and extremely fast steady-state wake solver for wind plant flows

Luis A Martínez-Tossas[1], Jennifer King[1], Eliot Quon[1], Christopher J Bay[1], Rafael Mudafort[1], Nicholas Hamilton[1], Michael F Howland[2], and Paul A Fleming[1]

[1]National Renewable Energy Laboratory, Golden, CO USA
[2]Graduate Aerospace Laboratories (GALCIT), California Institute of Technology, Pasadena, CA 91125, USA

**Correspondence:** luis.martinez@nrel.gov

**Abstract.**

Wind turbine wake models typically require approximations, such as wake superposition and deflection models, to accurately describe wake physics. However, capturing the phenomena of interest, such as the curled wake and interaction of multiple wakes, in wind power plant flows comes with an increased computational cost. To address this, we propose a new hybrid method that uses analytical solutions with an approximate form of the Reynolds-averaged Navier-Stokes equations to solve the time-averaged flow over a wind plant. We compare results from the solver to supervisory control and data acquisition data from the Lillgrund wind plant obtaining wake model predictions which are generally within one standard deviation of the mean power data. We perform simulations of flow over the Columbia River Gorge to demonstrate the capabilities of the model in complex terrain. We also apply the solver to a case with wake steering, which agreed well with large-eddy simulations. This new solver reduces the time–and therefore the related cost–it takes to simulate a steady-state wind plant flow (on the order of seconds using one core). Because the model is computationally efficient, it can also be used for different applications including wake steering for wind power plants and layout optimization.

## 1 Introduction

In this work, we present an improved formulation of the curled wake model (Martínez-Tossas et al., 2019) that can be used in the context of a wind power plant without the need to use a wake superposition method. Wake superposition models are typically used because of their computational efficiency; however, they have been shown to give different results depending on the model used (Gunn et al., 2016; Zong and Porté-Agel, 2020; Hamilton et al., 2020). This inconsistency motivates the use of a more robust solver in the context of the curled wake model (Martínez-Tossas et al., 2019) that does not depend on a superposition method. The new solver is developed by simplifying the Reynolds-averaged Navier-Stokes (RANS) equations to obtain a parabolic equation for the wake deficit. The equation is solved in a three-dimensional domain to obtain the wake velocity in a wind plant. This solver uses a hybrid RANS-analytical framework that aims to minimize computational cost while still preserving physics from the RANS equations.

Parabolic solvers for RANS equations are a promising tool for fast wind farm flow solvers. Ainslie (1988) developed a parabolic solver for an approximation of RANS equations in cylindrical coordinates. They proposed a mixing-length eddy-viscosity model that has a component from the ambient turbulence and another from the wake-added turbulence. Iungo et al. (2018) developed a parabolic RANS solver focused on improving the mixing-length model and used assumptions about axisymmetry in the wakes. Bradstock and Schlez (2019a, b) have developed a parabolic wind plant RANS solver (WakeBlaster) used for commercial applications. WakeBlaster solves a simplified version of the RANS equations and has been validated using field experiments (Bradstock and Schlez, 2019a). WakeBlaster uses a special method to solve the spanwise velocity components that does not include effects caused by yaw.

Wake steering is a promising wind plant control strategy used to maximize the power output of a wind plant (Adaramola and Krogstad, 2011; Park et al., 2013; Gebraad et al., 2016; Howland et al., 2019; Fleming et al., 2019; Siemens Gamesa, 2019). In wake steering, upstream turbines are yawed, deflecting the wakes such that downstream turbines are able to produce more power, and the wind plant as a whole can produce more power. In this work, we present a wind plant model that uses a simplified version of the RANS equations to predict the flow through a wind plant with wake steering. This tool is extremely fast (order of seconds), thereby enabling controls-oriented frameworks used for wind plant operation and layout optimization.

The wake of a wind turbine in yaw has a unique shape known as the curled wake (Howland et al., 2016; Bastankhah and Porté-Agel, 2016; Martínez-Tossas et al., 2019). This shape has been observed in computational fluid dynamics simulations and in small- and large-scale experiments (Medici and Alfredsson, 2006; Howland et al., 2016; Bastankhah and Porté-Agel, 2016; Vollmer et al., 2016; Bartl et al., 2018; Fleming et al., 2018b; Schottler et al., 2018). The curled wake is formed because the wake of a wind turbine in yaw introduces spanwise and vertical velocities that deform the wake and change its shape. This mechanism has been explained in the literature as a collection of vortices shed from the rotor plane (Howland et al., 2016; Bastankhah and Porté-Agel, 2016; Shapiro et al., 2018; Martínez-Tossas et al., 2019). The curled wake is a unique phenomenon in wind turbine wakes because it disrupts the asymmetry of the wake. As a result of the shed vortices, the curled yawed wind turbine wake is laterally asymmetric and non-Gaussian. The curled wake is known to affect not only a turbine immediately downstream, but also subsequent turbines within a wind plant. This effect is known as secondary steering and it is important to capture it when using wake models to unravel the full potential of wake steering (Fleming et al., 2018a; Bay et al., 2020; King et al., 2020).

The curled wake model uses a simplified version of the RANS equations to predict the wake of a wind turbine in yaw (Martínez-Tossas et al., 2019). Several improvements have been proposed to the original formulation of the model, including: 1) a decay model for the vortices, 2) tuning of the viscous term based on turbulence intensity, and 3) adding a pressure gradient term to account for wake expansion (Bay et al., 2019; Bay et al., 2020; Hulsman et al., 2020). Also, the new Gauss-Curl Hybrid model has shown to provide a good compromise between an analytical model (Bastankhah and Porté-Agel, 2016) and some of the physics from the curled wake model (King et al., 2020). The original formulation of the curled wake model was for a single wind turbine wake. In the case of a wind plant, the wakes are first computed individually, then superposed to obtain the flow field of the entire wind plant (Bay et al., 2020). Most wake models are used in the same manner by first computing the wake of the individual turbines and using a superposition method afterward to obtain the flow over the entire domain (Annoni

et al., 2018). This new curled wake solver overcomes the use of a superposition method by solving the flow over the entire wind plant. This allows us to realize the benefits of the curled wake model in a much faster time frame with better scaling with domain size.

The curled wake solver presented in this work focuses on reducing computational cost and capturing wake steering effects. This is done by solving only the streamwise component of the linearized RANS equations and parametrizing the effects of the spanwise and wall-normal components using semianalytical solutions.

## 2 Formulation

We use the RANS equations to model the time-averaged flow field through a wind plant. The continuity equation is

$$\frac{\partial \overline{u}}{\partial x} + \frac{\partial \overline{v}}{\partial y} + \frac{\partial \overline{w}}{\partial z} = 0, \tag{1}$$

and RANS momentum equation for the streamwise direction is:

$$\overline{u}\frac{\partial \overline{u}}{\partial x} + \overline{v}\frac{\partial \overline{u}}{\partial y} + \overline{w}\frac{\partial \overline{u}}{\partial z} = -\frac{1}{\rho}\frac{\partial \overline{p}}{\partial x} - \frac{\partial \overline{u'u'}}{\partial x} - \frac{\partial \overline{u'v'}}{\partial y} - \frac{\partial \overline{u'w'}}{\partial z} + \nu\left(\frac{\partial^2 \overline{u}}{\partial x^2} + \frac{\partial^2 \overline{u}}{\partial y^2} + \frac{\partial^2 \overline{u}}{\partial z^2}\right) + 2\Omega_y\left(G_z - \overline{w}\right) - 2\Omega_z\left(G_y - \overline{v}\right), \tag{2}$$

where $x$, $y$, and $z$ are the streamwise, spanwise, and wall-normal directions, $u$, $v$, and $w$ are the velocity components in the respective directions (with $'$ denoting time fluctuations and the overbar is time averaging); $\overline{p}$ is the time-averaged pressure; $\rho$ is density; $\Omega$ is the Earth's rotational vector projected into an arbitrary, non-inertial, Earth-fixed frame of reference

$$\Omega = \omega[\cos(\phi)\sin(\theta), \cos(\phi)\cos(\theta), \sin(\phi)]$$

where $\phi$ is the latitude, $\theta$ is the angle measured between the domain $x$-axis and the easting axis, and $\omega$ is the Earth's rotation rate; and $G$ is the geostrophic wind velocity vector. This is a similar equation used in the original formulation of the curled wake model, but now we focus on a new approach to derive the equations and some generalizations used for a wind plant approach as opposed to a single wind turbine wake. We assume that the viscous effects are small (high Reynolds number limit), and are neglected in the rest of derivation. We also assume that the boundary layer is neutral and it satisfies the geostrophic balance in the free atmosphere (Allaerts and Meyers, 2015; van der Laan and Sørensen, 2017). Coriolis effects are included in the momentum balance given by Equation 2 as a result of their influence on the wind speed and direction shear in the atmosphere (Wyngaard, 2010). Further, we included non-traditional Coriolis effects ($\Omega_y$) because of their potential impact in the atmospheric boundary layer in the presence of heterogeneous roughness elements such as wind turbines or terrain complexity (Howland et al., 2020a).

### 2.1 Decomposing the velocity

The velocity is decomposed into a background flow (capital letters) and a wake deficit ($\Delta$) by:

$$u = U + \Delta u, \qquad v = V + \Delta v, \qquad w = W + \Delta w, \qquad p = P + \Delta p. \tag{3}$$

The time-averaged fields are denoted using overbars:

$$\overline{u} = \overline{U} + \overline{\Delta u}, \qquad \overline{v} = \overline{V} + \overline{\Delta v}, \qquad \overline{w} = \overline{W} + \overline{\Delta w}, \qquad \overline{p} = \overline{P} + \overline{\Delta p}. \tag{4}$$

The temporal fluctuations are denoted using a hash mark ($'$):

$$u' = U' + \Delta u', \qquad v' = V' + \Delta v', \qquad w' = W' + \Delta w', \qquad p' = P' + \Delta p'. \tag{5}$$

### 2.1.1 Background flow

The background flow ($U$, $V$, $W$) is the velocity of the domain without including the wind turbines and their wakes. The background flow formulation can be obtained from an analytical formulation such as the log-law or from a different time-averaged simulation. For example, you can specify uniform flow by $U, V, W = U_\infty, 0, 0$, or use simulation data from LES or experiments to define the background flow over complex terrain. For a consistent formulation of the model, the background flow should also satisfy the RANS equations.

### 2.1.2 Wake deficit solution

The time-averaged wake velocities are denoted by $\overline{\Delta u}, \overline{\Delta v},$ and $\overline{\Delta w}$. We are interested in solving the streamwise component of the wake deficit, $\overline{\Delta u}$, while the other wake velocity components, $\overline{\Delta v}$ and $\overline{\Delta w}$, are parametrized using semianalytical models. The streamwise component of the RANS equations can be written in terms of the background flow and wake velocity as:

$$(\overline{U} + \overline{\Delta u})\frac{\partial(\overline{\Delta u} + \overline{U})}{\partial x} + (\overline{V} + \overline{\Delta v})\frac{\partial(\overline{U} + \overline{\Delta u})}{\partial y} + (\overline{W} + \overline{\Delta w})\frac{\partial(\overline{U} + \overline{\Delta u})}{\partial z} = -\frac{1}{\rho}\frac{\partial(P + \Delta p)}{\partial x}$$
$$- \frac{\partial\overline{(U' + \Delta u')(U' + \Delta u')}}{\partial x} - \frac{\partial\overline{(U' + \Delta u')(V' + \Delta v')}}{\partial y} - \frac{\partial\overline{(U' + \Delta u')(W' + \Delta w')}}{\partial z}$$
$$+ 2\Omega_y[G_z - (\overline{W} + \overline{\Delta w})] - 2\Omega_z[G_y - (\overline{V} + \overline{\Delta v})]. \tag{6}$$

The background flow is defined to also satisfy the RANS equations as

$$\overline{U}\frac{\partial\overline{U}}{\partial x} + \overline{V}\frac{\partial\overline{U}}{\partial y} + \overline{W}\frac{\partial\overline{U}}{\partial z} = -\frac{1}{\rho}\frac{\partial P}{\partial x} - \frac{\partial\overline{U'U'}}{\partial x} - \frac{\partial\overline{U'V'}}{\partial y} - \frac{\partial\overline{U'W'}}{\partial z} + 2\Omega_y\left(G_z - \overline{W}\right) - 2\Omega_z\left(G_y - \overline{V}\right). \tag{7}$$

Subtracting the background flow (Equation 7) from the full flow (Equation 6) leads to the equation of the curled wake model:

$$(\overline{U} + \overline{\Delta u})\frac{\partial\overline{\Delta u}}{\partial x} + (\overline{V} + \overline{\Delta v})\frac{\partial\overline{\Delta u}}{\partial y} + (\overline{W} + \overline{\Delta w})\frac{\partial\overline{\Delta u}}{\partial z} + \overline{\Delta u}\frac{\partial\overline{U}}{\partial x} + \overline{\Delta v}\frac{\partial\overline{U}}{\partial y} + \overline{\Delta w}\frac{\partial\overline{U}}{\partial z} = -\frac{1}{\rho}\frac{\partial\Delta p}{\partial x}$$
$$- \frac{\partial\overline{(2U'\Delta u' + \Delta u'\Delta u')}}{\partial x} - \frac{\partial\overline{(U'\Delta v' + V'\Delta v' + \Delta u'\Delta v')}}{\partial y} - \frac{\partial\overline{(U'\Delta w' - W'\Delta u' - \Delta u'\Delta w')}}{\partial z} + 2\left(\Omega_z\overline{\Delta v} - \Omega_y\overline{\Delta w}\right). \tag{8}$$

We now assume that the pressure gradient has a small effect (especially in the far wake), the Reynolds stresses are modeled using the turbulent-viscosity hypothesis (Pope, 2000), the second derivative of the wake deficit in the streamwise direction is neglected $\left(\frac{\partial^2\overline{\Delta u}}{\partial x^2} = 0\right)$, the gradients of the mean flow are assumed to be small, and their influence on the convective terms is neglected:

$$\overline{\Delta u}\frac{\partial\overline{U}}{\partial x} + \overline{\Delta v}\frac{\partial\overline{U}}{\partial y} + \overline{\Delta w}\frac{\partial\overline{U}}{\partial z} \ll (\overline{U} + \overline{\Delta u})\frac{\partial\overline{\Delta u}}{\partial x} + (\overline{V} + \overline{\Delta v})\frac{\partial\overline{\Delta u}}{\partial y} + (\overline{W} + \overline{\Delta w})\frac{\partial\overline{\Delta u}}{\partial z}, \tag{9}$$

the Coriolis terms in the velocity deficit equation are neglected (i.e. $2\Omega_z \overline{\Delta v} << (\overline{U} + \overline{\Delta u})\frac{\partial \overline{\Delta u}}{\partial x}$), but are included in the background flow RANS momentum balance (Equation 7). These assumptions lead to the final form of the equation:

$$\frac{\partial \overline{\Delta u}}{\partial x} = -\frac{1}{\overline{U} + \overline{\Delta u}}\left[(\overline{V} + \overline{\Delta v})\frac{\partial \overline{\Delta u}}{\partial y} + (\overline{W} + \overline{\Delta w})\frac{\partial \overline{\Delta u}}{\partial z} + \nu_{\text{eff}}\left(\frac{\partial^2 \overline{\Delta u}}{\partial y^2} + \frac{\partial^2 \overline{\Delta u}}{\partial z^2}\right)\right]. \tag{10}$$

Equation 10 is the fundamental parabolic equation solved in the model presented. The streamwise velocity deficit, $\overline{\Delta u}$, is the main unknown; all the other variables in the equation are either known a priori or parametrized at run-time depending on $\overline{\Delta u}$. We note that mass conservation was used in the derivation, however, it is not strictly enforced when solving Equation 10. The equation is solved by marching in the downstream direction starting from an initial condition where the first wind turbine is (section 3).

## 2.2 Turbulence model

The effect of turbulence in the RANS equations is described by the divergence of the Reynolds stress tensor. The streamwise component of the divergence of the Reynolds stress for the background flow solution (Equation 7) is:

$$\frac{\partial \overline{U'U'}}{\partial x} + \frac{\partial \overline{U'V'}}{\partial y} + \frac{\partial \overline{U'W'}}{\partial z}. \tag{11}$$

The Reynolds stress term in Equation 8 (for the wake deficit solution) is defined as:

$$\frac{\partial \overline{(2U'\Delta u' + \Delta u'\Delta u')}}{\partial x} + \frac{\partial \overline{(U'\Delta v' + V'\Delta v' + \Delta u'\Delta v')}}{\partial y} + \frac{\partial \overline{(U'\Delta w' + W'\Delta u' + \Delta u'\Delta w')}}{\partial z}. \tag{12}$$

The decomposition of the velocity field (background + wake, mean + fluctuation) leads to the introduction of additional stress-like terms in Equation 8. These terms are correlations between the background flow solution and the wake deficit solution.

A mixing length model is used to represent the terms in Equation 12. We propose using the simple model suggested in the original formulation of the curled wake model (Martínez-Tossas et al., 2019) and scale the viscosity to take into account the effect from all of the extra terms in the Reynolds stresses from Equation 12. This is the same approach suggested by Bay et al. (2019). The mixing length and eddy viscosity are defined as:

$$\ell_{\text{m}} = \frac{\kappa z}{(1 + \kappa z/\lambda)} \qquad \nu_{\text{eff}} = C\,\ell_{\text{m}}^2\left|\frac{d\overline{U}}{dz}\right| \tag{13}$$

where $\ell_{\text{m}}$ is the mixing length, $\nu_{\text{eff}}$ is the turbulent viscosity, $\kappa$ is the von Kármán constant, $z$ is the distance from the ground, and $\lambda$ is the value of the mixing length in the free atmosphere. Blackadar (1962) proposed that $\lambda \approx 0.00027G/f_c$, where G is the geostrophic wind speed magnitude and $f_c$ is the Coriolis parameter, resulting in a value of $\lambda$ which is latitude dependent. Using typical values of $G = 10$ [m/s] and mid-latitude $\phi = 45°$, $\lambda \approx 27$ [m]. The precise value of $\lambda$, and more broadly $\ell_{\text{m}}$, will depend on independent parameters such as the atmospheric stability (Sun, 2011). In this study, we select $\lambda = 27$ [m] and we suggest the investigation of more refined turbulence models for future work. The constant $C$ is used to account for the additional turbulence introduced by the rotor and the wake. Tests have shown that for all of the cases tried in the manuscript, a value of $C = 4$ has provided good agreement between the model and experiments/simulations. This value is consistent with

what is suggested by Bay et al. (2019). Appendix A shows a sensitivity analysis of power in a wind plant simulation using the model to the constant $C$. We also expand the equations for the turbulence model in Appendix B. The mixing length and turbulent viscosity are difficult to approximate with constant values that depend only on height ($z$). A better approximation would allow turbulent viscosity to vary spatially, especially in the wake, where the local turbulence varies with the spanwise and streamwise coordinates.

The Reynolds stress model used in this study was selected because of its computational efficiency. Resolving the spatial variations in the eddy viscosity would require the solution of the full RANS momentum equations and additional transport equations for relevant parameters in the selected Reynolds stress model (van der Laan et al., 2015a; Iungo et al., 2018). Future work should investigate Reynolds stress models that are able to resolve the enhanced mixing and turbulence induced by the wind turbines while remaining computationally efficient for the hybrid RANS-analytical framework.

## 2.3 Wind turbine wakes initial condition

Wakes are initialized according to the wind speed at the rotor location in the plane closest to where the turbine is. As the solution marches downstream and new wind turbines are encountered, a new wake deficit is added to the plane (n):

$$\overline{\Delta u}_{\mathrm{n}} = -2\,a\,\langle \overline{U + \Delta u} \rangle_{\mathrm{n}-1} \tag{14}$$

where $a = \left(1 - \sqrt{1 - C_T \cos^2 \alpha}\right)/2$ is the induction from momentum theory, $\alpha$ is the yaw angle, $C_T$ is the thrust coefficient, and $\langle \overline{U + \Delta u} \rangle_{\mathrm{n}-1}$ is the averaged velocity inside the disk in the plane upstream ($\mathrm{n}-1$) of the rotor. The power and thrust coefficients are obtained from a lookup table based on the local velocity $\langle \overline{U + \Delta u} \rangle_{\mathrm{n}-1}$. A Gaussian filter is used to smear the initial condition in the spanwise directions to avoid numerical instabilities described in Martínez-Tossas et al. (2019). The effects of wake curl, wake rotation, and the boundary layer are implemented using the analytical models also described in Martínez-Tossas et al. (2019). For completeness, we show the analytical formulas for the spanwise velocities from the curled wake. The effect of curl is added by modifying the spanwise velocity components according to an elliptic distribution of vorticity (Shapiro et al., 2018; Martínez-Tossas et al., 2019; Martínez-Tossas and Branlard, 2020). The spanwise velocities can be represented analytically by:

$$\overline{\Delta v} = \int_{-R}^{R} \frac{(z - z')}{2\pi \left(y^2 + (z - z')^2\right)} \left(1 - e^{-(y^2 + (z - z')^2)/\sigma^2}\right) \Gamma_0 \frac{z'}{R\sqrt{R^2 - z'^2}} dz' \tag{15}$$

$$\overline{\Delta w} = \int_{-R}^{R} \frac{-y}{2\pi \left(y^2 + (z - z')^2\right)} \left(1 - e^{-(y^2 + (z - z')^2)/\sigma^2}\right) \Gamma_0 \frac{z'}{R\sqrt{R^2 - z'^2}} dz'. \tag{16}$$

where $R$ is the turbine radius, $y$ and $z$ are the coordinates relative to the disk center, and $\Gamma_0 = \frac{D}{2} C_T U_\infty \sin \alpha \cos^2 \alpha$ is the total circulation from yaw (Shapiro et al., 2018; Martínez-Tossas et al., 2019). The power and thrust coefficients are computed using the tabulated value at zero yaw angle as follows:

$$P(\alpha) = P(\alpha = 0) \cos^2(\alpha), \quad T(\alpha) = T(\alpha = 0) \cos^2(\alpha). \tag{17}$$

This relation has been used in previous work, but field experimental studies indicate that these functions are not necessarily powers of cosines and can be turbine-specific (Howland et al., 2020b). The curled wake model presented here allows any function to be used to relate the power and thrust coefficient as a function of yaw angle and future work will be focused on

improving the functional relations between thrust, power, and yaw angle (e.g. model proposed by (Howland et al., 2020b)). The solver computes the power and thrust from each turbine according to the local velocity at the rotor plane.

## 3    Numerical Solution

Equation 10 is solved using numerical differentiation. Equation 18 shows the equation to be solved numerically with all of the terms labeled that are to be discretized:

$$
\overline{\Delta u}_{[i+1,j,k]} = \overline{\Delta u}_{[i,j,k]} - \overbrace{\frac{\Delta x}{\overline{U} + \overline{\Delta u}}}^{A} \left[ \overbrace{(\overline{V} + \overline{\Delta v})\frac{\partial \overline{\Delta u}}{\partial y}}^{B} + \overbrace{(\overline{W} + \overline{\Delta w})\frac{\partial \overline{\Delta u}}{\partial z}}^{C} + \nu_{\text{eff}} \left( \overbrace{\frac{\partial^2 \overline{\Delta u}}{\partial y^2}}^{D} + \overbrace{\frac{\partial^2 \overline{\Delta u}}{\partial z^2}}^{E} \right) \right].
\tag{18}
$$

The discrete form of Equation 18 is presented in Equation 19.

$$
\overline{\Delta u}_{[i+1,j,k]} = \overline{\Delta u}_{[i,j,k]} - \overbrace{\frac{\Delta x}{\overline{U} + \overline{\Delta u}}}^{A} \left[ \overbrace{\left(\overline{V}_{[i,j,k]} + \overline{\Delta v}_{[i,j,k]}\right)\frac{\overline{\Delta u}_{[i,j+1,k]} - \overline{\Delta u}_{[i,j-1,k]}}{2\Delta y}}^{B} + \right.
$$

$$
\overbrace{\left(\overline{W}_{[i,j,k]} + \overline{\Delta w}_{[i,j,k]}\right)\frac{\overline{\Delta u}_{[i,j,k+1]} - \overline{\Delta u}_{[i,j,k-1]}}{2\Delta z}}^{C} +
$$

$$
\left. \nu_{\text{eff}} \left( \overbrace{\frac{\overline{\Delta u}_{[i,j+1,k]} - 2\overline{\Delta u}_{[i,j,k]} + \overline{\Delta u}_{[i,j-1,k]}}{\Delta y^2}}^{D} + \overbrace{\frac{\overline{\Delta u}_{[i,j,k+1]} - 2\overline{\Delta u}_{[i,j,k]} + \overline{\Delta u}_{[i,j,k-1]}}{\Delta z^2}}^{E} \right) \right].
\tag{19}
$$

This numerical equation is discretized using a "forward-in-time centered-in-space" method with the stability criteria shown in Equation 20 (Hoffman and Frankel, 2018; Martínez-Tossas et al., 2019). We note that the model proposed is steady state and there is no time dependency. The spatial streamwise direction is treated as the "forward-in-time" part of the numerical method. The equations can be solved as a marching problem in the streamwise direction (index $i$) starting with an initial condition in a $yz$ plane. The boundary conditions are set to zero wake deficit ($\overline{\Delta u} = 0$):

$$
\Delta x \leq 2\nu_{\text{eff}} \frac{\overline{\Delta u}}{(\overline{W} + \overline{\Delta w})^2}, \quad \Delta y \geq \sqrt{2\nu_{\text{eff}}\Delta x / \overline{\Delta u}}.
\tag{20}
$$

Our tests have shown that the implementation has a converged and stable solution when using a grid resolution on the order of $\frac{D}{\Delta y} = 10$ in the spanwise directions ($y$ and $z$) and $\frac{D}{\Delta x} = 20$ in the streamwise direction. A grid convergence study is shown in Appendix C. All the simulations and results presented were performed using uniform grid spacing.

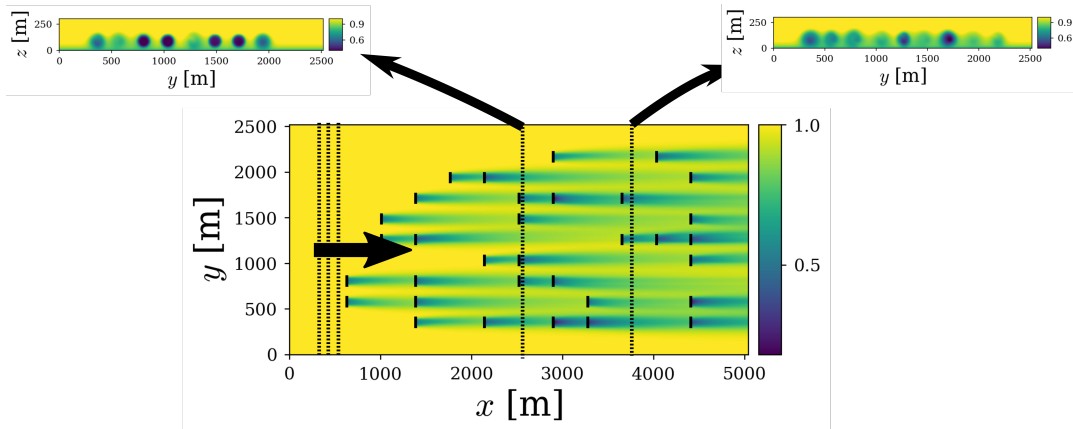

**Figure 1.** Schematic of the computational strategy used to solve equation 18. Dashed lines denote the location of a subset of planes, and big arrow shows the marching direction.

Figure 1 shows a schematic of how the solution is computed. The main figure is a contour of streamwise velocity from a simulation with a random arrangement of turbines. The solution is marched downstream by solving Equation 19 at each plane. Two planes are shown from the middle of the domain. The final solution includes a collection of planes for each streamwise location, which are combined to generate a full 3D solution.

### 3.1 Computational cost

To better understand the low computational cost of the solver presented, we asses the number of floating point operations needed to obtain a solution to Equation 16. We estimate the computational expense of the implementation by approximating the number of floating point operations (summation, subtraction, multiplication, division) in each term in Equation 19. We assume that the total number of grid points in the computational domain is $N$. To solve Equation 19, all the grid points in the domain must compute each of the terms in the equation. This leads to the following computational expense from each term: $A = 2N$, $B = 4N$, $C = 4N$, $D = 5N$, $E = 5N$ and, assuming one floating point operation between terms ($4N$), this leads to a total computational expense of $\approx 24N$ floating point operations. Assuming that we use a standard processor ($1-$Gflops), the computational time required for a simulation with $N = 100^3$ grid points based on this approximation would be $.02$ [s]. This can be considered an extremely fast solver for wind plant controls and layout optimization. In practice, the computational expense of the algorithm heavily depends on the implementation and software stack used. In our current implementation within the numpy and python frameworks (van der Walt et al., 2011), the typical computational cost of a simulation is on the order of 0.1-10 seconds. This is two order of magnitude faster than the standard curl model implementation in the FLOw Redirection and Induction in Steady State (FLORIS) framework. Figure 2 shows the time to solution of the algorithm as a function of total number of grid points from the model presented compared to the standard FLORIS implementation with wake superposition (Bay et al., 2019) compared to the linear scaling of the new solver. Also, the wind plant used for the scaling study

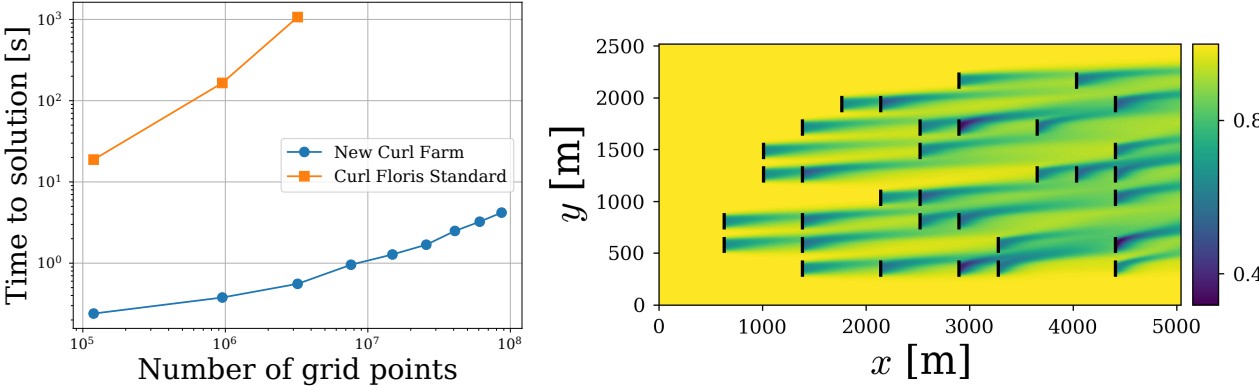

**Figure 2.** Scaling of the computational algorithm (left) based on a representative wind plant composed of 36 turbines with wake steering (right).

is shown for reference. Significant speedup is expected in the presently proposed curled wake model formulation compared to the standard FLORIS implementation. The standard FLORIS implementation solves Equation 16 for every turbine in the domain individually and then superposes the solutions. This superposition approach results in an increased computational cost, especially when more turbines are included, as well as wake superposition uncertainty. The resolutions used are finer than required for this wind plant, and the simulations lasting 0.5 seconds are converged and would be used for production runs. We note that this version of the model has not been optimized for performance and future work will include code optimization and shared memory parallelization.

## 4 Results

We use the model proposed to compare with three different cases. The first comparison is done using supervisory control and data acquisition (SCADA) data from the Lillgrund wind plant. Second, we showcase the use of the solver in complex terrain. Finally, we compare the model to a series of LES for an array of turbines with different yaw combinations.

### 4.1 Lillgrund Wind Plant

We use the model proposed to compute the flow field over the Lillgrund wind plant. Ten-minute average SCADA data is available for all turbines for different wind conditions. The SCADA data was organized by wind speed, turbulence intensity, and wind direction into bins with a width of 1 m/s, 2%, and 5°. Three conditions from directions where the meteorological tower is not waked were chosen (185° with 41 10-minute averages, 215° with 93 10-minute averages, and 255° with 90 10-minute averages). For each wind condition, we perform one simulation with the solver proposed. Figure 3 shows the layout of the Lillgrund Wind Plant with arrows denoting the directions for the cases that were studied. The background flow was set to a log-law streamwise velocity profile with a roughness height of $z_0 = 10^{-5}$[m].

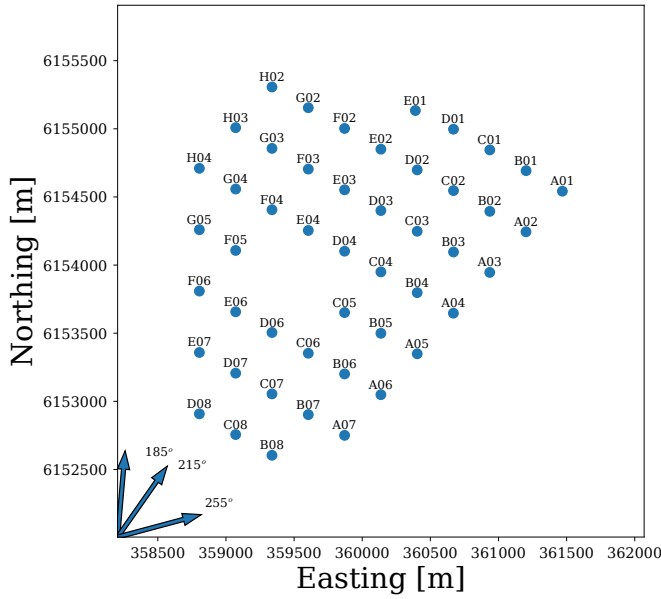

**Figure 3.** Layout of the Lillgrund Wind Plant with wind direction used in each simulation.

Figure 4 shows a comparison of power output between the SCADA data and the model proposed with a streamwise velocity contour at hub height. The data has been normalized according to the highest mean power in the experimental data. The bars in the SCADA data indicate the standard deviation of the power measurements. The mean absolute error in power is 8%, 10%, and 16% for the cases with 185°, 215°, and 255°, respectively. The agreement between the SCADA data and the model is good, with most results from the proposed solver lying within one standard deviation of the measurements. We can see different features of the flow, including the superposition of wakes. This allows for the solver to reach an equilibrium state in the deep array region. In this area, the power produced by the turbine flattens, providing a balance between the turbulent diffusion and the power extraction (Calaf et al., 2010). Future work will focus on including wind direction uncertainty in the curled wake model (Gaumond et al., 2014; van der Laan et al., 2015b; Simley et al., 2020).

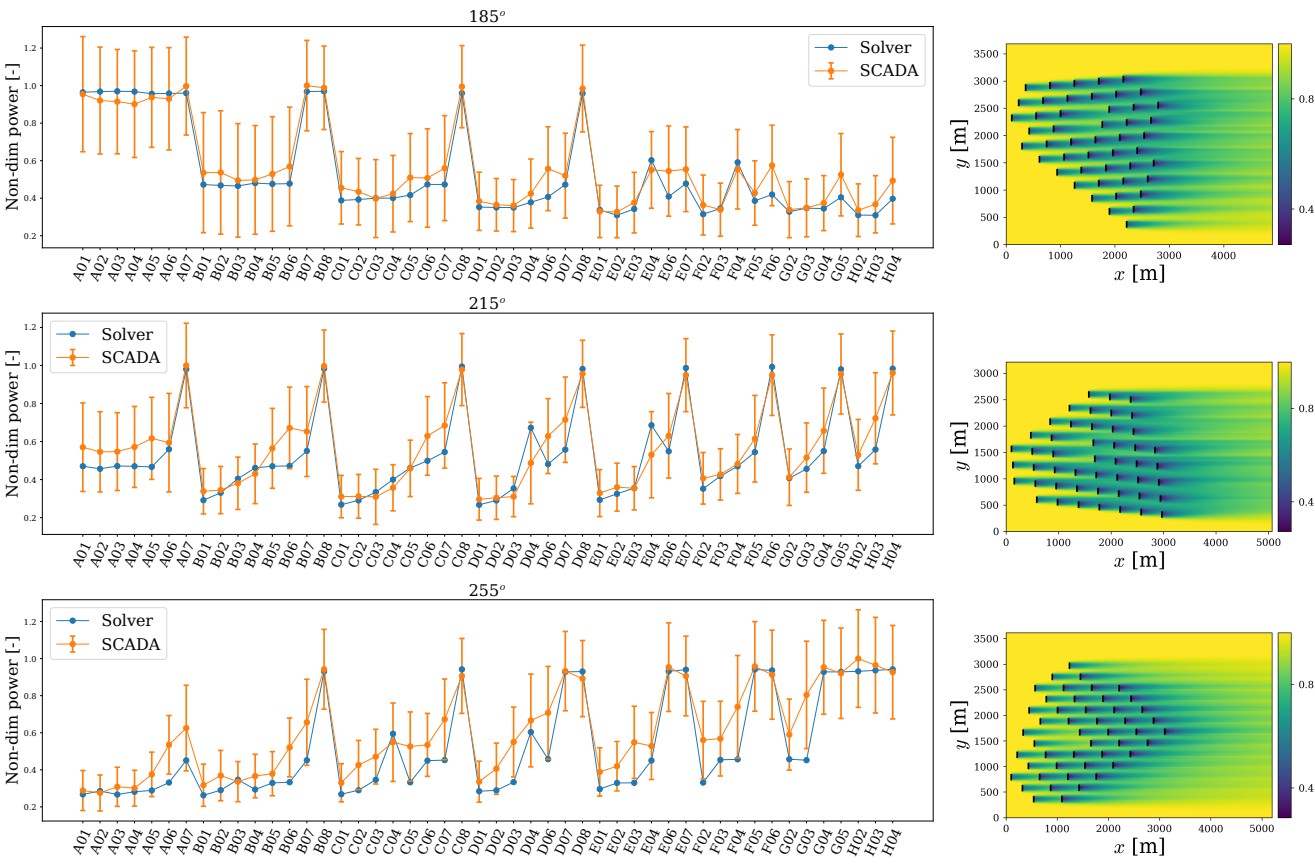

**Figure 4.** Comparison of turbine power versus SCADA data for the Lillgrund Wind Plant for cases at three different wind directions (185°, 215°, 255°). Streamwise velocity contours at hub height are shown for all cases with the wind plant aligned with the flow direction.

## 4.2 Complex terrain: Columbia River Gorge

We test the model presented on a case with complex terrain over the Columbia River Gorge (Quon et al., 2019). This test case is used to demonstrate the capabilities of the model in complex terrain conditions. The background flow solution is taken from a time-averaged LES (Quon et al., 2019). Figure 5 shows a volume rendering of streamwise velocity from a simulation using the proposed model. We can see the three-dimensionality of the solution and how the wakes conform to the terrain. The background flow is taken from LES, and the algorithm provides the solution for the wake deficits that would be present if turbines were there. Figure 6 shows streamwise velocity contours for planes in all directions. It is interesting to see how the wakes advect sideways following the background flow. Also, the combination of wakes leads to asymmetric deformation not typically observed in wakes over flat terrain. These results serve as a test case to show the applicability of the model in a case with complex terrain; further work is needed to assess the accuracy of the model under complex terrain conditions.

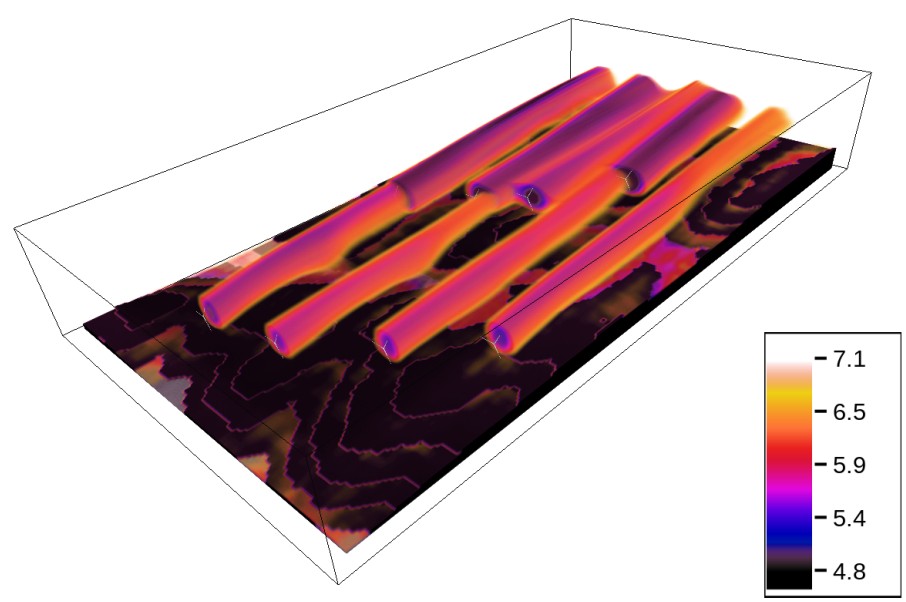

**Figure 5.** Volume rendering of streamwise velocity from a simulation using the proposed model. Image produced using Vapor (Li et al., 2019).

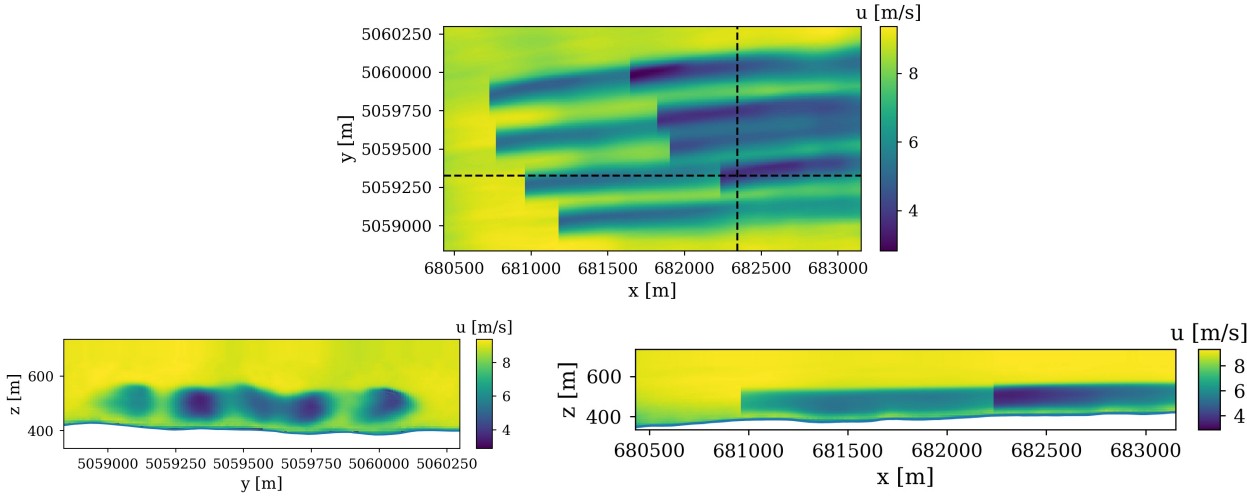

**Figure 6.** Streamwise velocity contours showing a plane perpendicular to the wall-normal direction (top) and the streamwise direction (bottom). The horizontal and vertical lines denote the location of the planes.

## 4.3 Wake steering

We now compare the model to results from LES of wakes in steering conditions. The simulations were performed using the Simulator fOr Applications (SOWFA) using an actuator disk model with rotation (Churchfield et al., 2012). The turbine aerodynamics properties and control system are derived from the NREL 5MW reference turbine (Jonkman et al., 2009). The simulations are for cases with wind plants of 4-by-3 and 3-by-3 turbines with different offsets and yaw-angle combinations. The cases with 4-by-3 have spacing in the streamwise and spanwise directions of $S_x = 10D$ and $S_y = 2.5D$ respectively. The cases with a 3-by-3 array have spacing in the streamwise and spanwise directions of $S_x = 10D$ and $S_y = 3D$ respectively. The simulations use a precursor simulation from a neutral atmospheric boundary layer with roughness height of $z_0 = 0.15[m]$, wind direction of $270°$, and wind speed at hub height ($90[m]$) of 8 [m/s]. The simulations are time-averaged over 1,600 [s]. Table 1 shows the main parameters for the simulations. The curled wake model uses the time-averaged LES inflow as the background flow.

Figure 7 shows the total power for each case from the model proposed and from LES. There is good agreement in total power between the model and the simulations. The model proposed is able to capture the effects of yaw and general trends of power output from the different configurations. We note that the simulations still have some transient effects and differences arise from these effects in the atmospheric boundary layer, including low-velocity streaks passing through the turbines (Munters et al., 2016; Stevens et al., 2018).

| Case | Number of Turbines | $S_x$ | $S_y$ | Hub-height velocity | Turbulence Intensity | Yaw Angles [°] |
|------|-------------------|-------|-------|--------------------|--------------------|----------------|
| 0 | 4-by-3 | 10 D | 3 D | 8 [m/s] | 10.0 % | -20, -20, -20, -20, 0, 0, 0, 0, 0, 0, 0, 0, |
| 1 | 4-by-3 | 10 D | 3 D | 8 [m/s] | 10.0 % | 0, 0, 0, 0, 0, 0, 0, 0, 0, 0, 0, 0, |
| 2 | 4-by-3 | 10 D | 3 D | 8 [m/s] | 10.0 % | 5, 10, 15, 20, 0, 0, 0, 0, 0, 0, 0, 0, |
| 3 | 4-by-3 | 10 D | 3 D | 8 [m/s] | 10.0 % | 10, 10, 10, 10, 0, 0, 0, 0, 0, 0, 0, 0, |
| 4 | 4-by-3 | 10 D | 3 D | 8 [m/s] | 10.0 % | 20, 15, 10, 5, 0, 0, 0, 0, 0, 0, 0, 0, |
| 5 | 4-by-3 | 10 D | 3 D | 8 [m/s] | 10.0 % | 20, 20, 20, 20, 0, 0, 0, 0, 0, 0, 0, 0, |
| 6 | 4-by-3 | 10 D | 3 D | 8 [m/s] | 10.0 % | 20, 20, 20, 20, 10, 10, 10, 10, 0, 0, 0, 0, |
| 7 | 3-by-3 | 10 D | 2.5 D | 8 [m/s] | 10.0 % | -20, -20, -20, 0, 0, 0, 0, 0, 0, |
| 8 | 3-by-3 | 10 D | 2.5 D | 8 [m/s] | 10.0 % | 20, 20, 20, 0, 0, 0, 0, 0, 0, |
| 9 | 3-by-3 | 10 D | 2.5 D | 8 [m/s] | 10.0 % | 20, 20, 20, 10, 10, 10, 0, 0, 0, |

**Table 1.** List of LES cases performed for comparison study.

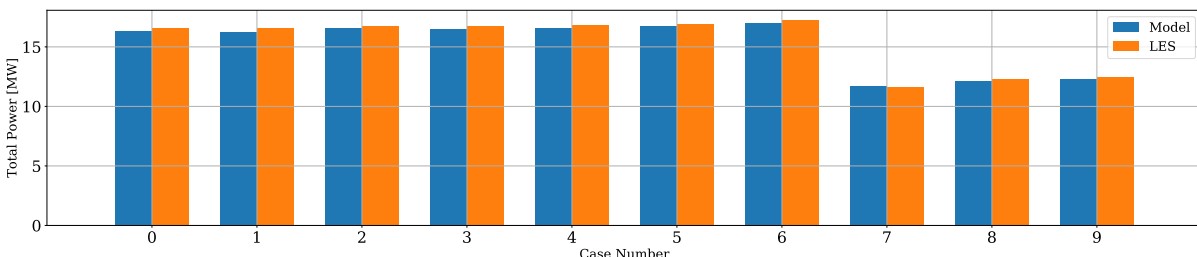

**Figure 7.** Total power output for wind plant LES with wake steering compared with the model proposed.

We select two representative cases and compare the power for all turbines and velocity at hub height. Figure 8 and Figure 9 show power for all turbines and a velocity profile at hub height from LES and the model for cases 4 and 9. The turbine power for each turbine from the model in all cases is always within one standard deviation of the plots. The streaks from the precursor simulation create some of the differences in turbine power on the first row. The streaks are long structures that persist in the domain for very long periods of time (Munters et al., 2016; Stevens et al., 2018). To take the streaks into account, they are included as part of the background solution in the model. We notice that there are differences in the near wake between the model and the LES. These differences are present because the representation of the near wake is not well captured in the model. A better representation of the near wake and a more sophisticated turbulence model that can take into account the wake-added turbulence will be part of future work.

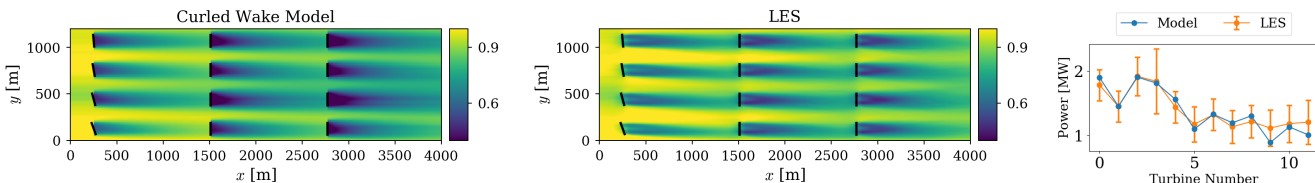

**Figure 8.** Velocity at hub height normalized by average speed at hub height from the model proposed and from LES and power output for each turbine from the model proposed compared to results from LES. Simulations of a 4-by-3 turbine array. The bars in the LES power denote one standard deviation of the power. Turbine numbering is from bottom to top and left to right.

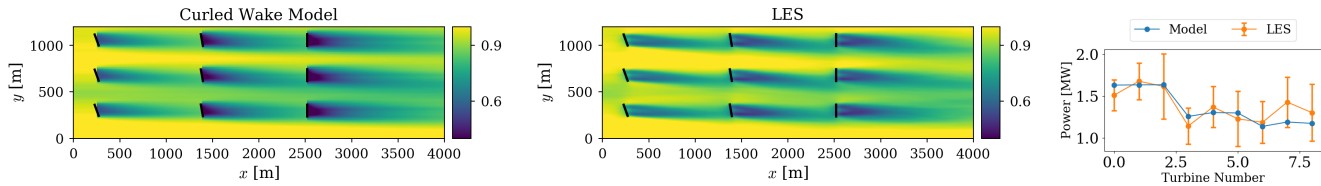

**Figure 9.** Same as Figure 9 but for a 4-by-3 array.

## 5 Conclusions

Fast wind power plant flow solvers are much needed for wind plant controls and layout optimization. In this work, we presented a simplified and fast solver for wind turbine wakes based on the curled wake model presented in Martínez-Tossas et al. (2019). The approach uses a hybrid RANS-analytical framework to obtain the wake velocity based on a parabolic solution for the streamwise component of the RANS equations. The computational expense of the model was shown to be on the order of seconds for a full wind plant with 36 turbines. The model was tested on three different cases: 1) SCADA data from the Lillgrund Wind Plant 2) LES for flow over complex terrain, and 3) LES over flat terrain with different yaw-angle combinations. The models showed good agreement with the SCADA data from the Lillgrund Wind Plant. The model was also able to generate wake profiles for data in complex terrain and future work will focus on comparing these profiles to data. Finally, the solver was able to reproduce the trends from LES with different yaw combinations. The model presented was shown to be an extremely fast solver (order of seconds) for wind turbine wakes with terrain features. This was achieved by simplifying the streamwise component of the RANS equation and making a series of assumptions. This model leverages approximations, especially with regard to the turbulence model, to improve computational speed. This trade-off provides a very computationally efficient solver at the expense of less robust turbulence modeling, compared to full three dimensional RANS solver (van der Laan et al., 2015a; Iungo et al., 2018).

Some of the limitations from the different approximations of the model include a turbulence model mixing length that only depends on the vertical coordinate, a linearized solution of the vortices from curl that do not decay, a missing near-wake formulation, and no pressure term in the equations. These approximations were done to reduce the computational cost. Future work will focus on addressing the limitations and more specifically, comparing the model with RANS, improving the turbulence model without compromising computational cost, improving the near wake, implementing a vortex decay model, using the solver for yaw-angle optimizations in a wind plant, and improving code performance to increase speed. This solver will soon be incorporated into the FLORIS framework and will be freely available.

## Appendix A: The constant of the turbulence model

The turbulence model proposed uses a constant, $C$, to scale the turbulent viscosity. This constant is used to represent the wake-added turbulence. We performed a series of simulations with different values of $C$ to tune the model constant. Figure A1 shows the SCADA power from the Lillgrund wind farm compared to results from the curled wake solver using a different value of the turbulent viscosity scaling, $C$. A value of $C = 4$ provided best agreement between the curled wake model and Lillgrund SCADA data. We also note that this value agrees with previous observations from Bay et al. (2019).

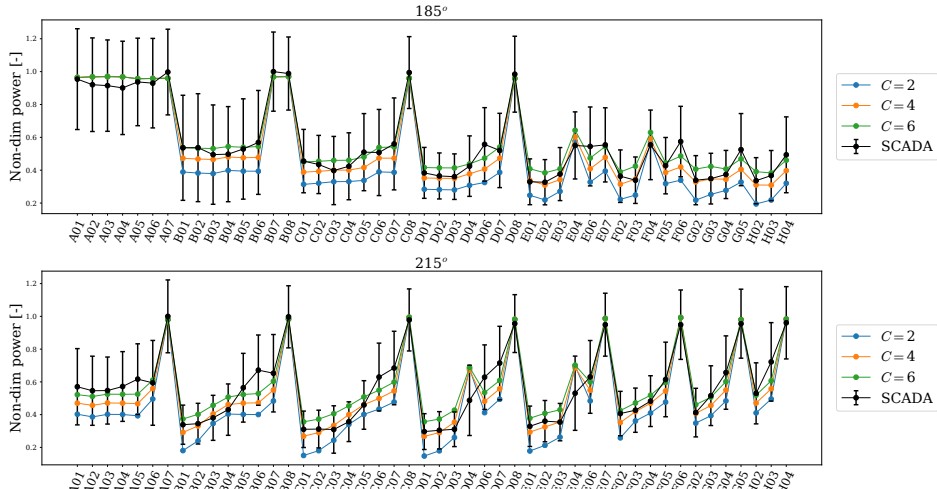

**Figure A1.** Comparison of turbine power versus SCADA data for the Lillgrund Wind Plant for cases at two different wind directions ($185°$, $215°$). Different lines denote values of the constant $C$ used to scale the viscous term in the solver.

## Appendix B: Turbulence modeling

In this section, we show a different formulation for the development of the turbulence model. It is possible to invoke the eddy-viscosity-hypothesis in the derivation for the base and wake deficit equations independently. When doing this, we express the Reynolds stresses as:

$$-\frac{\partial \overline{(2U'\Delta u' + \Delta u'\Delta u')}}{\partial x} - \frac{\partial \overline{(U'\Delta v' + V'\Delta v' + \Delta u'\Delta v')}}{\partial y} - \frac{\partial \overline{(U'\Delta w' - W'\Delta u' - \Delta u'\Delta w')}}{\partial z} = \tag{B1}$$

$$(\nu_f - \nu_b)\left(\frac{\partial^2 \overline{U}}{\partial x^2} + \frac{\partial^2 \overline{U}}{\partial y^2} + \frac{\partial^2 \overline{U}}{\partial z^2}\right) + \nu_f\left(\frac{\partial^2 \overline{\Delta u}}{\partial x^2} + \frac{\partial^2 \overline{\Delta u}}{\partial y^2} + \frac{\partial^2 \overline{\Delta u}}{\partial z^2}\right) \tag{B2}$$

where $\nu_b$ is the turbulent viscosity of the base flow and $\nu_f$ is the turbulent viscosity of the full equation. We now choose to represent the Reynolds stresses as a function of the gradients of the wake deficit solution.

$$\nu_{\text{eff}}\left(\frac{\partial^2 \overline{\Delta u}}{\partial x^2} + \frac{\partial^2 \overline{\Delta u}}{\partial y^2} + \frac{\partial^2 \overline{\Delta u}}{\partial z^2}\right) = (\nu_f - \nu_b)\left(\frac{\partial^2 \overline{U}}{\partial x^2} + \frac{\partial^2 \overline{U}}{\partial y^2} + \frac{\partial^2 \overline{U}}{\partial z^2}\right) + \nu_f\left(\frac{\partial^2 \overline{\Delta u}}{\partial x^2} + \frac{\partial^2 \overline{\Delta u}}{\partial y^2} + \frac{\partial^2 \overline{\Delta u}}{\partial z^2}\right) \tag{B3}$$

where $\nu_{\text{eff}}$ is an effective viscosity if we only use the gradients of the wake deficit. Re-arranging the equation, the effective turbulent viscosity can be defined as

$$\nu_{\text{eff}} = (\nu_f - \nu_b)\frac{\left(\frac{\partial^2 \overline{U}}{\partial x^2} + \frac{\partial^2 \overline{U}}{\partial y^2} + \frac{\partial^2 \overline{U}}{\partial z^2}\right)}{\left(\frac{\partial^2 \overline{\Delta u}}{\partial x^2} + \frac{\partial^2 \overline{\Delta u}}{\partial y^2} + \frac{\partial^2 \overline{\Delta u}}{\partial z^2}\right)} + \nu_f. \tag{B4}$$

We can define the effective viscosity as:

$$\nu_{\text{eff}} = C\nu_b, \tag{B5}$$

and subtracting from Equation B4 leads to:

$$C = (\nu_f/\nu_b - 1)\frac{\left(\frac{\partial^2 \overline{U}}{\partial x^2} + \frac{\partial^2 \overline{U}}{\partial y^2} + \frac{\partial^2 \overline{U}}{\partial z^2}\right)}{\left(\frac{\partial^2 \overline{\Delta u}}{\partial x^2} + \frac{\partial^2 \overline{\Delta u}}{\partial y^2} + \frac{\partial^2 \overline{\Delta u}}{\partial z^2}\right)} + \nu_f/\nu_b. \tag{B6}$$

The value of $C$ is a function of space. However, in this work, we have chosen a constant value for $C$ that minimizes the error between the observations and the model results.

## Appendix C: Grid refinement study of the curled wake model

Here, we show a convergence study for the curled wake model based on one of the simulations for the Lillgrund wind farm in
subsection 4.1. We evaluate grid convergence using power output from turbines. The power is computed by taking the velocity average in the rotor area for each turbine and using a lookup table. Figure C1 shows the power for all turbines and each line represents a different resolution in the spanwise directions. The results converge when using $D/\Delta y$=9. At this point, the average error percentage in power for the finest resolution is below 3%. We also refined the streamwise direction for all cases studied, but we noticed that the error in power from refining the grid in the streamwise direction is always less than 1% as long
as the grid meets the stability criteria presented in section 3.

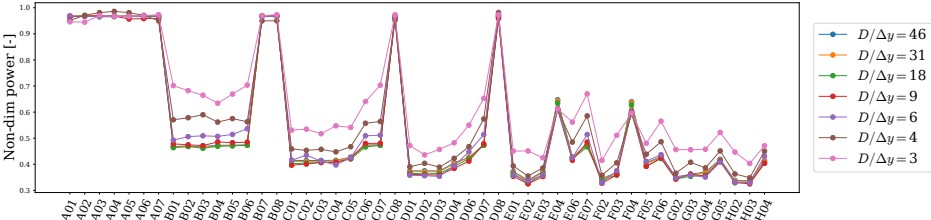

**Figure C1.** Comparison of turbine power for all turbines using different number of grid points across the turbine diameter for the Lillgrund wind plant for cases at wind direction of $185°$.

*Code availability.* The code will soon be available withing the FLORIS framework.

*Author contributions.* LAMT led the model development and writing the article. All authors provided input to this manuscript.

*Competing interests.* No competing interest.

## Appendix D: Acknowledgments

The authors would like to thank Paula Doubrawa for help in allocating funding for this work and Sheri Anstedt from NREL for their help in editing the manuscript. The authors would also like to acknowledge the comments from Paul van der Laan and the anonymous reviewer. Data was furnished to the authors under an agreement between the National Renewable Energy Laboratory, Siemens Gamesa Renewable Energy A/S, and Vattenfall. Data and results used herein do not reflect findings by Siemens Gamesa Renewable Energy A/S and Vattenfall. A portion of the research was performed using computational resources spon-

sored by the U.S. Department of Energy's Office of Energy Efficiency and Renewable Energy and located at the National Renewable Energy Laboratory. This work was authored by the National Renewable Energy Laboratory, operated by Alliance for Sustainable Energy, LLC, for the U.S. Department of Energy (DOE) under Contract No. DE-AC36-08GO28308. Funding provided by the U.S. Department of Energy Office of Energy Efficiency and Renewable Energy Wind Energy Technologies Office. The views expressed in the article do not necessarily represent the views of the DOE or the U.S. Government. The

U.S. Government retains and the publisher, by accepting the article for publication, acknowledges that the U.S. Government retains a nonexclusive, paid-up, irrevocable, worldwide license to publish or reproduce the published form of this work, or allow others to do so, for U.S. Government purposes.

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
