# Peer review of "The curled wake model: A three-dimensional and extremely fast steady-state wake solver for wind plant flows"

_Wind Energy Science, 2020_

## Referee Comment (RC1) · Paul van der Laan (Referee) · 5 Aug 2020

**Review of *The curled wake model: A three-dimensional and extremely fast steady-state wake solver for wind plant flows* by Luis A Martínez-Tossas et al.**

Reviewer: M. Paul van der Laan, DTU Wind Energy

The authors employ an existing parabolic Reynolds-averaged Navier-Stokes (RANS) model of a single wind turbine wake including wake steering and they extend the model to 3D wind farm flows. A model derivation is presented and results of three test cases are discussed using field measurement and large-eddy simulations. The article is well written and provide interesting content. However, I do lack a model verification in the form of a grid refinement study and technical information regarding the model setup is missing. The validation study is interesting but could be improved with quantitative statements about the actual differences. I have listed main and minor comments below, which should be addressed in order to accept the article as a publication for Wind Energy Science.

**Main comments**

1. It is nice that you provide a derivation of the model. I have derived the model following your steps, but I lack some information to arrive at the final form (Equation 9):

   (a) Equation 5: What is $\Delta p_w$? Should it be $\Delta p$?

   (b) Equation 7: In order to arrive at this equation one also needs to neglect the viscous term, which you forgot to mention.

   (c) Section 2.2: You forgot to mention that you use the Boussinesq hypothesis for the Reynolds-stress:

   $$\overline{u_i' u_j'} = \frac{2}{3} k \delta_{ij} - \nu_T \left( \frac{\partial \overline{u}_i}{\partial x_j} + \frac{\partial \overline{u}_j}{\partial x_i} \right)$$

   The turbulent kinetic energy $k$ could be absorbed in the pressure terms that you are neglecting, so we can write:

   $$\overline{u'u'} = -2\nu_T \frac{\partial u}{\partial x}, \quad \overline{u'v'} = -\nu_T \left( \frac{\partial \overline{u}}{\partial y} + \frac{\partial \overline{v}}{\partial x} \right), \quad \overline{u'w'} = -\nu_T \left( \frac{\partial \overline{u}}{\partial z} + \frac{\partial \overline{w}}{\partial x} \right)$$

   Following your proposed decomposition we get:

   $$\overline{(U' + \Delta u')(U' + \Delta u')} = -2\nu_T \frac{\partial \left( \overline{U} + \overline{\Delta u} \right)}{\partial x},$$

   $$\overline{(U' + \Delta u')(V' + \Delta v')} = -\nu_T \left( \frac{\partial \left( \overline{U} + \overline{\Delta u} \right)}{\partial y} + \frac{\partial \left( \overline{V} + \overline{\Delta v} \right)}{\partial x} \right),$$

   $$\overline{(U' + \Delta u')(W' + \Delta w')} = -\nu_T \left( \frac{\partial \left( \overline{U} + \overline{\Delta u} \right)}{\partial z} + \frac{\partial \left( \overline{W} + \overline{\Delta w} \right)}{\partial x} \right)$$

Subtracting the mean U-momentum contributions by assuming that the Boussinesq hypothesis holds for the mean background flow we obtain:

$$\overline{(2U'\Delta u' + \Delta u'\Delta u')} = -2\nu_T \frac{\partial \overline{\Delta u}}{\partial x},$$

$$\overline{U'\Delta v' + V'\Delta u' + \Delta u'\Delta v'} = -\nu_T \left( \frac{\partial \overline{\Delta u}}{\partial y} + \frac{\partial \overline{\Delta v}}{\partial x} \right),$$

$$\overline{U'\Delta w' + W'\Delta u' + \Delta u'\Delta w'} = -\nu_T \left( \frac{\partial \overline{\Delta u}}{\partial z} + \frac{\partial \overline{\Delta w}}{\partial x} \right),$$

(d) In addition, when substituting the above in Equation 7 of the article and using Equation 8 we get:

$$\frac{\partial \overline{\Delta u}}{\partial x} = -\frac{1}{\overline{U} + \overline{\Delta u}} \left[ (\overline{V} + \overline{\Delta w}) \frac{\partial \overline{\Delta u}}{\partial y} + (\overline{W} + \overline{\Delta w}) \frac{\partial \overline{\Delta u}}{\partial z} \right.$$
$$\left. + 2\nu_T \frac{\partial^2 \overline{\Delta u}}{\partial x^2} + \nu_T \left( \frac{\partial^2 \overline{\Delta u}}{\partial y^2} + \frac{\partial^2 \overline{\Delta v}}{\partial x \partial y} \right) + \nu_T \left( \frac{\partial^2 \overline{\Delta u}}{\partial z^2} + \frac{\partial^2 \overline{\Delta w}}{\partial x \partial z} \right) \right]$$

This is not the same as the final result presented in Equation 7 of the article, where the terms related to the streamwise gradients seem to be neglected:

$$2\frac{\partial^2 \overline{\Delta u}}{\partial x^2} + \frac{\partial^2 \overline{\Delta v}}{\partial x \partial y} + \frac{\partial^2 \overline{\Delta w}}{\partial x \partial z} = 0$$

Please clarify. If my derivation is correct and this assumption for the Reynolds-stresses is indeed required, then you should add the assumption to the article.

2. Equation 12: When you introduce an additional constant in the effective eddy viscosity, you are actually scaling your mixing length scale by a factor $\sqrt{C}$ and you can write an effective mixing length in the same form as proposed by Blackadar using an effective von Kármán constant and an effective maximum length scale:

$$\ell_{m,\text{eff}} \equiv \ell_m \sqrt{C} = \frac{\kappa z \sqrt{C}}{1 + \frac{\sqrt{C}\kappa z}{\sqrt{C}\lambda}} = \frac{\kappa^* z}{1 + \frac{\kappa^* z}{\lambda^*}},$$

with $\kappa^* = \sqrt{C}\kappa$ and $\lambda^* = \sqrt{C}\lambda$. Hence, the constant $C$ is not an independent constant. If you use $C = 8$, then $\kappa^* = 2.83\kappa$, which seems very high to me. In addition, the maximum length scale $\lambda$ also represent a boundary layer height that needs to be adjusted for each flow case, see for example van der Laan et al. (2020), where a similar limited-length-scale turbulence closure is discussed.

3. Section 2.1.1: Here you mention that any background flow can be chosen. However, you do state that the background flow should also satisfy the RANS equation of $U$-momentum (Equation 6), so this is a requirement of the background flow that is worth to mention explicitly. For example, using experimental field data could have included effects of Coriolis and/or atmospheric stability that you are not considering in Equation 6.

4. How do you model the thrust coefficient for yawed cases? Do you use an analytical relation between the thrust coefficient and the yaw angle?

5. Have you performed a grid refinement study to verify the model? You mention an order of grid spacing for x and y to obtain numerical stability in Section 3 but how does the flow solution behave with grid refinement? This is an important question that should be addressed in order to accept the article publication. This also applies to the vertical spacing (z).

6. What is the actual chosen grid spacing and how large are the domain dimensions? Is the grid spacing uniform or is there also grid stretching?

7. You use a logarithmic inflow in your model representing a neutral atmospheric surface layer. At the same time you apply a mixing length model with a maximum set length scale, representing an idealized (stable) atmospheric boundary layer. This combination does not make sense to me. I would either use a logarithmic inflow with a mixing length that represent the atmospheric surface layer ($\kappa z$) or I would use an idealized atmospheric boundary layer velocity profile with the mixing length profile including a maximum set value.

8. The computational effort in order of seconds for a single flow case is impressive. However, a wind farm layout/control optimizer would most likely use AEP in the objective function and calculating the AEP would require in order of $10^3$-$10^4$ flow cases. Hence, the presented model is still quite expensive for an optimization process where the AEP is required for each iteration.

9. How large is the wind direction bin used to bin the SCADA? If this bin is small, i.e. $5°$, have you considered applying wind direction uncertainty as a Gaussian filter of several models results for different wind directions? See for example Gaumond et al. (2014) and van der Laan et al. (2015).

10. It would be more fair to plot the standard error of the mean as error bars in Figure 4, so the standard deviation normalized by the square root of the number of bin samples.

11. Section 4.3: I lack information on how this test case was performed:

    (a) Which wind turbine was used, NREL-5MW?

    (b) What was the wind direction, $270°$?

    (c) What was the wind farm layout (or spacing) for each wind farm case?

12. The article could benefit from more quantitative statements when validating the model. How large are the difference with either measurements or LES? What would these differences mean in terms of AEP?

13. Figure 8: The difference between individual wind turbine powers are actually quite large and I would not consider this a good agreement. This does not mean that your model is not performing well, but it could be that the LES model has not reached a steady-state as you also point out.

14. It would make sense to me to compare the model with an elliptic RANS AD model, which would be a more fair comparison with respect to using LES. You could consider this for future work.

**Minor comments**

1. Line 140: Equation 1 should be Equation 7.

**References**

Gaumond, M., Réthoré, P.-E., Ott, S., Peña, A., Bechmann, A., and Hansen, K. S.: Evaluation of the wind direction uncertainty and its impact on wake modeling at the Horns Rev offshore wind farm, Wind Energy, 17, 1169, 2014.

van der Laan, M. P., Sørensen, N. N., Réthoré, P.-E., Mann, J., Kelly, M. C., Troldborg, N., Hansen, K. S., and Murcia, J. P.: The $k$-$\varepsilon$-$f_P$ model applied to wind farms, Wind Energy, 18, 2065, https://doi.org/10.1002/we.1804, 2015.

van der Laan, M. P., Kelly, M., Floors, R., and Peña, A.: Rossby number similarity of an atmospheric RANS model using limited-length-scale turbulence closures extended to unstable stratification, Wind Energy Science, 5, 355–374, https://doi.org/10.5194/wes-5-355-2020, https://wes.copernicus.org/articles/5/355/2020/, 2020.

---

## Referee Comment (RC2) · Anonymous Referee #2 · 6 Aug 2020

**Review on the manuscript wes-2020-86, entitled "The curled wake model: A three-dimensional and extremely fast steady-state wake solver for wind plant flows", by L.A. Martinez-Tossas et al.**

This manuscript deals with further development of the curled wake model by proposing a parabolic solution of the governing equation that allows achieving low computational costs, which is a feature highly sought for wind energy practitioners.

The computational capabilities of the proposed model are highly compelling, yet the description of the model and its assessment can be improved. Furthermore, the overall quality of the manuscript should be improved as well.

My main comments are:
- The authors should state how this model in its core differs from the Ainslie model (Ainslie, J.F., Calculating the flow field in the wake of wind turbines, *J. Wind Eng. Industr. Aerodyn.*, 27, 213-224, 1988). In the Ainslie model, pressure is neglected, the RANS equations are solved parabolically, turbulent stresses are modeled with a mixing-length assumption (actually that model is slightly more complex including a component for ambient turbulence and a component for wake-generated turbulence), in analogy with the proposed model, which is still valuable considering the addition of the velocity perturbation induced by the rotor yaw (Shapiro *et al.* 2018).
- Besides the efforts made by the authors to develop an analytical framework for this model, this model should be considered as a semi-empirical model. The rough approximations used (removing forcing and adding directly the respective velocity perturbations, neglecting the pressure gradients, rough eddy-viscosity modeling) lead to flow predictions not satisfying basic first-principles of fluid dynamics, such as conservation of mass. This is particularly evident if considering null yaw of the wind turbine. If we consider that foundational models, such as the Jensen model, were developed only using the mass conservation, then I think it is reasonable to ask if the accuracy achieved relies only on the "smart" tuning of the mixing length model, which is difficult to generalize (see below more comments on the mixing length model).

These comments should be addressed in a revised version of the manuscript. Below you can find more comments that I hope can help with the preparation of a revised manuscript.

**Comments**
1. Eq. 1: Where is the forcing of the wind turbine? The turbulent Reynold stresses have the wrong sign. I hope this is only a typo in the manuscript rather than a bug in the code!
2. Eq. 4 is a kind of tricky because: $\overline{a'} = \overline{A'} + \overline{\Delta a'} = 0$, for the Reynolds averaging. Thus, $\overline{A'} = -\overline{\Delta a'}$. Does it make any sense that the mean fluctuations of the background flow are equal and opposite to that of the wake deficit? Please add comments on this, which might help to understand better this modeling strategy.
3. Eq. 5, what is $p_w$, $\Delta p$?

4. For Eq. 5 from Eq. 1, you should state that you are neglecting the molecular viscosity.

5. Line 89, there is no mixing length model in Eq. 9 so far, maybe an eddy viscosity model.

6. L95, Eq 9 is not parabolic, maybe It can be solved parabolically.

7. As I mentioned above in my main comments, I am not sure if it makes sense to build up these equations to then neglect the partial derivatives of the background velocity field, pressure, and proposing to model turbulent fluxes through an "ad-hoc" eddy viscosity model. The author should discuss this in the manuscript.

8. In Eq. 9 for the eddy-viscosity modeling of the turbulent Reynolds stresses, I think you have two options: a) you practically neglect what you wrote in Eq. 7 and you use what you have in Eq. 9 ($\nu_{eff}\left(\frac{\partial^2 \overline{\Delta u}}{\partial y^2} + \frac{\partial^2 \overline{\Delta u}}{\partial z^2}\right)$) saying that this is an ad-hoc modeling based on the physics, indeed you expect that the main contribution to turbulent fluxes is due to the turbulence connected the wake shear; b) you write all the equations of the turbulent Reynolds stresses with the eddy viscosity assumption and you add the other terms that are missing.

9. Eq 12: I am not sure this specific mixing length model makes sense for several reasons. First, multiplying the mixing length by $C$ means that you have an effective mixing length of $\sqrt{c}\,l_m$, which can create issues with the model derived from the Monin-Obukhov similarity theory that you have in Eq. 12. I think you should reconsider this approach by adding to the shear-generated turbulence, contributions due to ambient turbulence (atmospheric stability), and wake generated turbulence (Ainslie 1988, Iungo et al. 2018). Furthermore, you should consider rewriting the contraction of the strain-rate tensor including both background flow and wake deficit, and you will find other contributions you are missing in the mixing length model.

10. Eq. 13, you can report the explicit formulation $\overline{\Delta u} = -\frac{2a\overline{U}}{1+2a}$

11. L114, provide a reference for the mixing length in the free atmosphere equal to 15 m.

12. Eqs. 14 and 15, I guess rather than $v'$ and $w'$, they should be $\overline{\Delta v}$ and $\overline{\Delta w}$. Again, more comments on the consequences of replacing the turbine forcing with a velocity perturbation on the momentum and mass budgets might be helpful.

13. Eqs. 14 and 15 seem different from what reported in Martinez-Tossas 2019, please cross-check.

14. Eq. 17. Cross-check the finite-difference scheme, e.g. there is a second-order approximation of the first derivative, so it should be divided by $2\Delta y$ and $2\Delta z$.

15. Eq. 17. Provide the final parabolic equation solved in the code.

16. L 145, forward in time? Maybe forward in the streamwise direction. Please do not mention time to avoid confusion.

17. L152 – how this grid resolution is obtained? Have you done any study on grid sensitivity?

18. Sect. 4.1, How did you set the thrust coefficient of each turbine? Likewise for Sect. 4.2.

19. Sect. 4.3, I recommend providing an assessment at the turbine level, considering the data availability from LES.

20. Fig. 8, add the turbine numbers in the color maps. The LES data might be questionable, considering the difference in power for the turbines at the first row. Is there any specific reason, rather than numerical issues? Is there a better LES/RANS dataset to use for this assessment?

21. L224, you can say parabolic solution of the streamwise momentum equation of the RANS.

---

## Author Comment (AC1) · 14 Oct 2020

**Review of *The curled wake model: A three-dimensional and extremely fast steady-state wake solver for wind plant flows* by Luis A Martínez-Tossas et al.**

Reviewer: M. Paul van der Laan, DTU Wind Energy

The authors employ an existing parabolic Reynolds-averaged Navier-Stokes (RANS) model of a single wind turbine wake including wake steering and they extend the model to 3D wind farm flows. A model derivation is presented and results of three test cases are discussed using field measurement and large-eddy simulations. The article is well written and provide interesting content. However, I do lack a model verification in the form of a grid refinement study and technical information regarding the model setup is missing. The validation study is interesting but could be improved with quantitative statements about the actual differences. I have listed main and minor comments below, which should be addressed in order to accept the article as a publication for Wind Energy Science.

*The authors would like to thank the reviewer Paul van der Laan for the excellent review and attention to detail in all the formulations. We have added an appendix with a grid refinement study and expanded the information about the model setup. We added more quantitative comparisons to the manuscript. The responses to all comments are below marked in blue.*

Main comments

1. It is nice that you provide a derivation of the model. I have derived the model following your steps, but I lack some information to arrive at the final form (Equation 9):

   (a) Equation 5: What is $\Delta p_w$? Should it be $\Delta p$?

      *Yes, this was a typo and has been fixed in the manuscript.*

   (b) Equation 7: In order to arrive at this equation one also needs to neglect the viscous term, which you forgot to mention.

      *Yes, we neglect the viscous term in the formulation, and this has been included in the manuscript:*

      *"We assume that the viscous effects are small (high Reynolds number limit), and are neglected in the rest of derivation."*

   (c) Section 2.2: You forgot to mention that you use the Boussinesq hypothesis for the Reynolds-stress:
   $$\overline{u_i' u_j'} = \frac{2}{3} k \delta_{ij} - \nu_T \left( \frac{\partial \overline{u}_i}{\partial x_j} + \frac{\partial \overline{u}_j}{\partial x_i} \right)$$

      The turbulent kinetic energy $k$ could be absorbed in the pressure terms that you are neglecting, so we can write:

$$\overline{u'u'} = -2\nu_T \frac{\partial u}{\partial x}, \qquad \overline{u'v'} = -\nu_T \left( \frac{\partial \overline{u}}{\partial y} + \frac{\partial \overline{v}}{\partial x} \right), \qquad \overline{u'w'} = -\nu_T \left( \frac{\partial \overline{u}}{\partial z} + \frac{\partial \overline{w}}{\partial x} \right)$$

Following your proposed decomposition we get:

$$\overline{(U' + \Delta u')(U' + \Delta u')} = -2\nu_T \frac{\partial \left( \overline{U} + \overline{\Delta u} \right)}{\partial x},$$

$$\overline{(U' + \Delta u')(V' + \Delta v')} = -\nu_T \left( \frac{\partial \left( \overline{U} + \overline{\Delta u} \right)}{\partial y} + \frac{\partial \left( \overline{V} + \overline{\Delta v} \right)}{\partial x} \right),$$

$$\overline{(U' + \Delta u')(W' + \Delta w')} = -\nu_T \left( \frac{\partial \left( \overline{U} + \overline{\Delta u} \right)}{\partial z} + \frac{\partial \left( \overline{W} + \overline{\Delta w} \right)}{\partial x} \right) !$$

Subtracting the mean U-momentum contributions by assuming that the Boussinesq hypothesis holds for the mean background flow we obtain:

$$\overline{(2U'\Delta u' + \Delta u'\Delta u')} = -2\nu_T \frac{\partial \overline{\Delta u}}{\partial x},$$

$$\overline{U'\Delta v' + V'\Delta u' + \Delta u'\Delta v'} = -\nu_T \left( \frac{\partial \overline{\Delta u}}{\partial y} + \frac{\partial \overline{\Delta v}}{\partial x} \right),$$

$$\overline{U'\Delta w' + W'\Delta u' + \Delta u'\Delta w'} = -\nu_T \left( \frac{\partial \overline{\Delta u}}{\partial z} + \frac{\partial \overline{\Delta w}}{\partial x} \right),$$

The author's response is included in the response to comment 1. d) below.

(d) In addition, when substituting the above in Equation 7 of the article and using Equation 8 we get:
$$\frac{\partial \overline{\Delta u}}{\partial x} = -\frac{1}{\overline{U} + \overline{\Delta u}} \left[ \left( \overline{V} + \overline{\Delta w} \right) \frac{\partial \overline{\Delta u}}{\partial y} + \left( \overline{W} + \overline{\Delta w} \right) \frac{\partial \overline{\Delta u}}{\partial z} \right.$$
$$\left. + 2\nu_T \frac{\partial^2 \overline{\Delta u}}{\partial x^2} + \nu_T \left( \frac{\partial^2 \overline{\Delta u}}{\partial y^2} + \frac{\partial^2 \overline{\Delta v}}{\partial x \partial y} \right) + \nu_T \left( \frac{\partial^2 \overline{\Delta u}}{\partial z^2} + \frac{\partial^2 \overline{\Delta w}}{\partial x \partial z} \right) \right]$$

This is not the same as the final result presented in Equation 7 of the article, where the terms related to the streamwise gradients seem to be neglected:
$$2\frac{\partial^2 \overline{\Delta u}}{\partial x^2} + \frac{\partial^2 \overline{\Delta v}}{\partial x \partial y} + \frac{\partial^2 \overline{\Delta w}}{\partial x \partial z} = 0$$

Please clarify. If my derivation is correct and this assumption for the Reynolds-stresses is indeed required, then you should add the assumption to the article.

Yes, the derivation is correct. The term can be written as

$$2\frac{\partial^2 \Delta u}{\partial x^2} + \frac{\partial^2 \Delta v}{\partial x \partial y} + \frac{\partial^2 \Delta w}{\partial x \partial z} = \frac{\partial}{\partial x} \left( \frac{\partial \Delta u}{\partial x} + \frac{\partial \Delta v}{\partial y} + \frac{\partial \Delta w}{\partial z} \right) + \frac{\partial^2 \Delta u}{\partial x^2}$$

Using the continuity equation $\frac{\partial \Delta u}{\partial x} + \frac{\partial \Delta v}{\partial y} + \frac{\partial \Delta w}{\partial z} = 0$, the remaining term is $\frac{\partial^2 \Delta u}{\partial x^2}$. This means that there is only one term in the equation that has been neglected. This term was neglected by assuming that it was small and

it has been included in the derivations. We have also included the turbulent eddy viscosity hypothesis in the manuscript:

"the Reynolds stresses are modeled using the turbulent-viscosity hypothesis"

We have attempted different approaches for the derivations. The goal of this work is to take into account the turbulent stresses through the use of an effective turbulent viscosity and the gradients of the wake deficit solution only. A new appendix has been added with a re-formulation that invokes the turbulent-viscosity hypothesis for the base and wake deficit solutions individually.

2. Equation 12: When you introduce an additional constant in the effective eddy viscosity, you are actually scaling your mixing length scale by a factor $\overline{C}$ and you can write an effective mixing length in the same form as proposed by Blackadar using an effective von Kármán constant and an effective maximum length scale:

$$\ell_{\mathrm{m,eff}} \equiv \ell_{\mathrm{m}}\sqrt{C} = \frac{\kappa z \sqrt{C}}{1 + \frac{\sqrt{C}\kappa z}{\sqrt{C}\lambda}} = \frac{\kappa^* z}{1 + \frac{\kappa^* z}{\lambda^*}},$$

with $\kappa^* = \sqrt{C}\kappa$ and $\lambda^* = \sqrt{C}\lambda$. Hence, the constant $C$ is not an independent constant. If you use $C = 8$, then $\kappa^* = 2.83\kappa$, which seems very high to me. In addition, the maximum length scale $\lambda$ also represent a boundary layer height that needs to be adjusted for each flow case, see for example van der Laan et al. (2020), where a similar limited-lengthscale turbulence closure is discussed.

The reviewer is correct in pointing out that the scaling factor used in the formulation of the mixing length has a significant impact on the value of the eddy viscosity and the diffusion represented in the parabolic equations that are solved. In the current work, the model assumes that the mixing length from the ABL can be used directly, and the effects of the aggregate additional turbulent diffusion from the wake are represented by the constant correction factor $C$. This approach has the effect of increasing the turbulent diffusion, which is required to reach a sensible wake recovery rate. An alternative approach to this problem would require that the mixing length take into account the mean flow gradients introduced in the wake. Regarding the maximum length scale $\lambda$, the current formulation of the curled wake model calculates the mixing length using gradients in the mean ABL flow. In the formulation used here, $\lambda$ is a constant that is representative of the maximum mixing length in the free atmosphere. No additional formulation is included to vary the mixing length with boundary layer height or atmospheric stability.

3. Section 2.1.1: Here you mention that any background flow can be chosen. However, you do state that the background flow should also satisfy the RANS equation of $U$-momentum (Equation 6), so this is a requirement of the background flow that is worth to mention explicitly. For example, using experimental field data could have included effects of Coriolis and/or atmospheric stability that you are not considering in Equation 6.

Yes, the background flow should also satisfy the RANS equations. We agree that the Coriolis term should be included in the equations. We have re-derived the equations and included the Coriolis term to have a more complete description of the flow.

4. How do you model the thrust coefficient for yawed cases? Do you use an analytical relation between the thrust coefficient and the yaw angle?

We use a relation of cosine squared for the thrust and power coefficients. This has been added to the text:

"The power and thrust coefficients are computed using the tabulated value at zero yaw angle using Equation 17. This relation has been used in previous work, but new research indicates that these functions are not necessarily powers of cosines and can be turbine specific. The model presented allows any function to be used to relate the power and thrust coefficient as a function of yaw angle and future work will be focused on improving the functional relations between thrust, power and yaw angle."

5. Have you performed a grid refinement study to verify the model? You mention an order of grid spacing for x and y to obtain numerical stability in Section 3 but how does the flow solution behave with grid refinement? This is an important question that should be addressed in order to accept the article publication. This also applies to the vertical spacing (z).

Yes, we have performed a grid refinement study and have added to the manuscript in the appendix. This statement has also been added to the text:

"Our tests have shown that the implementation has a converged and stable solution when using a grid resolution on the order of $\frac{D}{\Delta y}$ 10 in the spanwise directions ($y$ and $z$) and $\frac{D}{\Delta x}$ 20 in the streamwise direction. A grid convergence study is shown in Appendix A.

6. What is the actual chosen grid spacing and how large are the domain dimensions? Is the grid spacing uniform or is there also grid stretching?

We use a grid spacing in all simulations that satisfies the conditions from the grid resolution study. Yes, the grid spacing is uniform. We have added the following statement in the manuscript:

"All the simulations and results presented were performed using uniform grid spacing."

7. You use a logarithmic inflow in your model representing a neutral atmospheric surface layer. At the same time you apply a mixing length model with a maximum set length scale, representing an idealized (stable) atmospheric boundary layer. This combination does not make sense to me. I would either use a logarithmic inflow with a mixing length that represent the atmospheric surface layer ($\kappa z$) or I would use an idealized atmospheric boundary layer velocity profile with the mixing length profile including a maximum set value.

Yes, we use a logarithmic inflow to represent the atmospheric boundary layer. The model used (Blackadar 1962) does what the reviewer suggests. The model used behaves as $\kappa z$ near the ground and has a maximum value of $\lambda$=15m.

8. The computational effort in order of seconds for a single flow case is impressive. However, a wind farm layout/control optimizer would most likely use AEP in the objective function and calculating the AEP would require in order of $10^3$-$10^4$ flow cases. Hence, the presented model is still quite expensive for an optimization process where the AEP is required for each iteration.

Yes, we agree that it is possible to make the solver faster. The current implementation has not been optimized for performance, but based on the analysis in section "3.1 Computational Cost" the order N of computational cost would allow for optimization and shared memory parallelization. We have included these statements in the manuscript:

Section 3.1:

"We note that this version of the model has not been optimized for performance and future work will include code optimization and shared memory parallelization."

Conclusions:

"Future work will focus on comparing the model with RANS, improving the turbulence model without compromising computational cost, improving the near wake, implementing a vortex decay model, using the solver for yaw-angle optimizations in a wind plant, and improving code performance to increase speed."

9. How large is the wind direction bin used to bin the SCADA? If this bin is small, i.e. 5°, have you considered applying wind direction uncertainty as a Gaussian filter of several models results for different wind directions? See for example Gaumond et al. (2014) and van der Laan et al. (2015).

All operational data from the Lillgrund wind plant was organized by wind speed, turbulence intensity, and wind direction into bins of width 1 m/s, 2%, and 5°, respectively. Organizing operational data in this way collects similar observations into subsets and supports easy comparison with model data. The authors are aware of the wind direction uncertainty studies suggested by the reviewer, as well as recent work by Simley et al. (2020). However, the work presented in the current manuscript focuses on the development of the wind plant solver. Uncertainty propagation and quantification studies for the curled wake model will be undertaken in subsequent work. We have added the following sentence in the manuscript:

"The SCADA was organized by wind speed, turbulence intensity, and wind direction into bins of width 1 m/s, 2 \%, and 5$^o$."

"Future work will focus on including wind direction uncertainty in the curled wake model \citep{Gaumond2014,van2015b,simley2020design}"

10. It would be more fair to plot the standard error of the mean as error bars in Figure 4, so the standard deviation normalized by the square root of the number of bin samples.

We agree that there are other ways to plot the data. We prefer to show standard deviation to show the range of the experimental data. The results are not intended to show the uncertainty in the data, but more the range of operation. Here is one of the plots with the standard error for reference.

[Figure]

11. Section 4.3: I lack information on how this test case was performed:

(a) Which wind turbine was used, NREL-5MW?

> Yes, this is the NREL 5MW. The following statement has been added to the manuscript:
>
> "The turbine aerodynamics properties and control system are derived from the NREL 5MW reference turbine."

(b) What was the wind direction, 270°?

> Yes, the wind direction is 270 and the following sentence has been modified to include this: The simulations use a precursor simulation from a neutral atmospheric boundary layer with roughness height of $z_0=0.15$[m], wind direction of 270$^o$ and wind speed at hub height (90[m]) of 8 [m/s].

(c) What was the wind farm layout (or spacing) for each wind farm case?

> The spacing for the cases with 4-by-4 turbines was 10D and 2.5D in the streamwise and spanwise directions and 10D and 3D for the cases with 4-by-3 turbines. We have updated the table with the cases to include the spacing.

12. The article could benefit from more quantitative statements when validating the model. How large are the difference with either measurements or LES? What would these differences mean in terms of AEP?

> We agree that quantitative statements are needed to improve the comparison of the model with data. We have reported errors for some of the results in the manuscript. The main objective of the paper is to present this new hybrid RANS-analytical framework for wind turbine wakes. Future work will focus on thorough comparisons with uncertainty quantification for AEP calculations.

13. Figure 8: The difference between individual wind turbine powers are actually quite large and I would not consider this a good agreement. This does not mean that your model is not performing well, but it could be that the LES model has not reached a steady-state as you also point out.

> We agree with this statement. The precursor simulation used for the LES had streaks that persist for a long time. We have used the LES precursor with the streaks as the background flow solution to the model. Now the results from the model agree much better with the LES. We also included standard deviations in the power plots to show the range of power from the LES.

14. It would make sense to me to compare the model with an elliptic RANS AD model, which would be a more fair comparison with respect to using LES. You could consider this for future work.

> Yes, we agree with the reviewer. We have compared with LES and SCADA because it is what we have available and offer the highest fidelity for model validation. We are planning on comparing the model to RANS simulations and improving the formulation based on RANS. This has been added to the conclusions:
>
> "Future work will focus on comparing the model with RANS, improving the turbulence model without compromising computational cost, implementing a vortex decay model, using the solver for yaw-angle optimizations in a wind plant, and improving code performance to increase speed."

Minor comments

1. Line 140: Equation 1 should be Equation 7.

> Thank you, this has been corrected.

References

Gaumond, M., Réthoré, P.-E., Ott, S., Peña, A., Bechmann, A., and Hansen, K. S.: Evaluation of the wind direction uncertainty and its impact on wake modeling at the Horns Rev offshore wind farm, Wind Energy, 17, 1169, 2014.

van der Laan, M. P., Sørensen, N. N., Réthoré, P.-E., Mann, J., Kelly, M. C., Troldborg, N., Hansen, K. S., and Murcia, J. P.: The $k$-$\varepsilon$-$f_P$ model applied to wind farms, Wind Energy, 18, 2065, https://doi.org/10.1002/we.1804, 2015.

van der Laan, M. P., Kelly, M., Floors, R., and Peña, A.: Rossby number similarity of an atmospheric RANS model using limited-lengthscale turbulence closures extended to unstable stratification, Wind Energy Science, 5, 355–374, https://doi.org/10.5194/wes-5-355-2020, https://wes.copernicus.org/articles/5/355/2020/, 2020.

---

## Author Comment (AC2) · 14 Oct 2020

**Review on the manuscript wes-2020-86, entitled "The curled wake model: A three dimensional and extremely fast steady-state wake solver for wind plant flows", by L.A. Martinez-Tossas et al.**

This manuscript deals with further development of the curled wake model by proposing a parabolic solution of the governing equation that allows achieving low computational costs, which is a feature highly sought for wind energy practitioners.

The computational capabilities of the proposed model are highly compelling, yet the description of the model and its assessment can be improved. Furthermore, the overall quality of the manuscript should be improved as well.

We thank the reviewer for the positive feedback. We have addressed all the comments from the reviewer and modified the manuscript accordingly. The responses to the review are marked in blue.

My main comments are:
• The authors should state how this model in its core differs from the Ainslie model (Ainslie, J.F., Calculating the flow field in the wake of wind turbines, *J. Wind Eng. Industr. Aerodyn.,* 27, 213-224, 1988). In the Ainslie model, pressure is neglected, the RANS equations are solved parabolically, turbulent stresses are modeled with a mixing-length assumption (actually that model is slightly more complex including a component for ambient turbulence and a component for wake-generated turbulence), in analogy with the proposed model, which is still valuable considering the addition of the velocity perturbation induced by the rotor yaw (Shapiro *et al.* 2018).

We thank the reviewer for pointing out this work. The work of Ainslie focuses on a cylindrical form of the equations with a model for one of the components of the Reynolds stress tensor. The wake is also assumed to be axisymmetric and there is no treatment of the wake in yawed conditions. In this work, we emphasize on the curled wake, a new derivation of the Reynolds stress terms and propose a different approach for the turbulence model. There are some similarities between the Ainslie model and the one proposed in this work, and there are also significant differences. We have included this work in the introduction:

"Ainslie (1988) developed a parabolic solver for an approximation of RANS equations in cylindrical coordinates. They proposed a mixing length eddy viscosity model that has a component from the ambient turbulence and another from the wake added turbulence."

• Besides the efforts made by the authors to develop an analytical framework for this model, this model should be considered as a semi-empirical model. The rough approximations used (removing forcing and adding directly the respective velocity perturbations, neglecting the pressure gradients, rough eddy-viscosity modeling) lead to flow predictions not satisfying basic first-principles of fluid dynamics, such as

conservation of mass. This is particularly evident if considering null yaw of the wind turbine. If we consider that foundational models, such as the Jensen model, were developed only using the mass conservation, then I think it is reasonable to ask if the accuracy achieved relies only on the "smart" tuning of the mixing length model, which is difficult to generalize (see below more comments on the mixing length model).

We agree that this model is not analytical. We are proposing a hybrid RANS-analytical model that is focused on minimizing computational cost. This model aims to achieve a computational cost as low as the analytical models but solving a simplified form of the streamwise momentum RANS equation. We have identified this as a hybrid RANS-analytical framework in the manuscript. The main purpose behind a RANS-analytical framework is to minimize computational cost. The tuning of the turbulence model is an essential part of minimizing cost. We have expanded the discussion of the turbulence model and included a section in the appendix with the effects of the tuning parameter in the turbulence model.

Introduction:
"This solver uses a hybrid RANS-analytical framework that aims to minimize computational cost."

2.3 Turbulence model
"Future work should investigate Reynolds stress models which are able to resolve the enhanced mixing and turbulence induced by the wind turbines while remaining computationally efficient for the hybrid RANS-analytical framework."

Conclusions:
"The approach uses a hybrid RANS-analytical framework to obtain the wake velocity based on a parabolic equation for the streamwise component of the RANS equations."

Mass conservation is used in the derivation of the model. However, when solving the momentum equation, mass conservation is not strictly enforced. This model solves an approximate form of the streamwise momentum equation. To be able to have a mass conserving approach, we would need to solve the three components of velocity and an equation for pressure which would be elliptic. This would require a full solution of the RANS equations and cannot be used as a fast model.

These comments should be addressed in a revised version of the manuscript. Below you can find more comments that I hope can help with the preparation of a revised manuscript.

**Comments**
1. Eq. 1: Where is the forcing of the wind turbine? The turbulent Reynold stresses have the wrong sign. I hope this is only a typo in the manuscript rather than a bug in the code! 2.

Yes, the sign of the Reynolds stresses was wrong. This was just an error in the manuscript, and it has been fixed. The turbulence model used to represent the Reynolds stresses in the code had the correct sign, so there were no bugs associated to this typo. A wrong sign in the viscous term would make the numerical method unstable.

2. Eq. 4 is a kind of tricky because: $a\#' = A\#' + \Delta((a((' = 0$, for the Reynolds averaging. Thus, $A\#' = -(\Delta(a(('$. Does it make any sense that the mean fluctuations of the background flow are equal and opposite to that of the wake deficit? Please add comments on this, which might help to understand better this modeling strategy.

   In the case of averaging, the fluctuations do average to zero. The Reynolds averaging is applied to all terms in the equation. This means that every term in the equation is zero. The mean fluctuations of the background flow are all zero. All the interactions between fluctuation are taken into account by the Reynolds stress tensor. The discussion of the Reynolds stress tensor and turbulence model have been expanded in the manuscript.

3. Eq. 5, what is $p_!$, $\Delta p$?

   Yes, this is a typo and has been fixed in the manuscript.

4. For Eq. 5 from Eq. 1, you should state that you are neglecting the molecular viscosity.

   Yes, we neglect the viscous term in the formulation and this has been included in the manuscript:

   "We assume that the viscous effects are small (high Reynolds number limit) and are neglected in the rest of derivation."

5. Line 89, there is no mixing length model in Eq. 9 so far, maybe an eddy viscosity model.

   Yes, the turbulent eddy viscosity approach was used to model the Reynolds stress tensor. The following text has been added to the manuscript:

   "The Reynolds stresses are modeled using the turbulent-viscosity hypothesis (Pope 2020) and the streamwise gradient of the wake deficit is neglected."

6. L95, Eq 9 is not parabolic, maybe It can be solved parabolically.

   The assumptions used to derive the curled wake model equation led to a convection-diffusion equation. The information propagates in the streamwise direction. By neglecting the second derivative of the velocity in the streamwise direction ($\frac{\partial^2 \Delta u}{\partial x^2}$) the equation becomes parabolic.

7. As I mentioned above in my main comments, I am not sure if it makes sense to build up these equations to then neglect the partial derivatives of the background velocity field, pressure, and proposing to model turbulent fluxes through an "ad-hoc" eddy viscosity model. The author should discuss this in the manuscript.

Building the equations is an important step in understanding the flow throughout the wind plant and the effects of the background flow and the wake deficit separately. The eddy viscosity model is necessary to represent unresolved terms in the equations. We have expanded the discussion of the turbulence model choice and have included an appendix showing details of the eddy viscosity model.

8. In Eq. 9 for the eddy-viscosity modeling of the turbulent Reynolds stresses, I think you have two options: a) you practically neglect what you wrote in Eq. 7 and you use what you have in Eq. $\overline{9\,(\nu_{\#\#}} - \frac{\$}{\$}\overline{\Delta''}'\&' + \frac{\$}{\$})'\Delta''\&'$.) saying that this is an ad-hoc modeling based on the physics, indeed you expect that the main contribution to turbulent fluxes is due to the turbulence connected the wake shear; b) you write all the equations of the turbulent Reynolds stresses with the eddy viscosity assumption and you add the other terms that are missing.

This point indicates that the explanation for the implementation of the eddy viscosity model was not sufficiently clear. The formulation of the eddy viscosity in the current work does indeed implement an ad hoc model that assumes the stress-like terms found in equation 7 can be represented by way of a mixing length model and a factor that accounts for the additional turbulent diffusion introduced by the rotor and the mean gradients in the wake. A more complete formulation would relate of the stress-like terms to the eddy viscosity (or eddy viscosities), which would require some additional information or different assumptions about the nature of the correlations between background flow and wake flow components. Related the point (9) raised below, the goal of the current work is to develop a parabolic model for wind turbine wake flows that can be solved by marching downstream. The mixing length approach used here does not require that any of the local gradients be calculated in order to estimate the eddy viscosity. Instead, the model assumes that the mixing length from the ABL can be used, and the effects of the aggregate additional turbulent diffusion from the wake are represented by the constant correction factor. A statement to this effect has been added to the manuscript:

"The Reynolds stress model used in the present study was selected due to its computational efficiency. Resolving the spatial variations in the eddy viscosity would require the solution of the full RANS momentum equations and additional transport equations for relevant parameters in the selected Reynolds stress model (van der Laan et al., 2015; Iungo et al., 2018). Future work should investigate Reynolds stress models which

are able to resolve the enhanced mixing and turbulence induced by the wind turbines while remaining computationally efficient for the hybrid RANS-analytical framework."

9. Eq $\overline{12}$: I am not sure this specific mixing length model makes sense for several reasons. First, multiplying the mixing length by $C$ means that you have an effective mixing length of $\sqrt{c}l_*$, which can create issues with the model derived from the Monin-Obukhov similarity theory that you have in Eq. 12. I think you should reconsider this approach by adding to the shear-generated turbulence, contributions due to ambient turbulence (atmospheric stability), and wake generated turbulence (Ainslie 1988, Iungo et al. 2018). Furthermore, you should consider rewriting the contraction of the strain-rate tensor including both background flow and wake deficit, and you will find other contributions you are missing in the mixing length model.

The implementation of the mixing length model in the current work is taken without additional modifications that would account for wake-added turbulence and shear generated turbulence, as noted by the reviewer. There is obvious value to reevaluating the canonical formulation of the mixing length for the atmospheric boundary layer when considering other sources of turbulence. The work by Ainslie and Iungo et al., attest to that. However, the curled wake model is a parabolic model that fits between the levels of fidelity seen in analytical wake models and RANS modeling. Multiplying the mixing length by a constant coefficient in the current work is a simple way of saying that the aggregate addition of shear-generated and wake-added turbulence is to increase the effective velocity by a fixed amount. The constant $C$ was the value that minimized the difference between the wind plant power predicted with the curled wake wind farm solver and the observed data. That said, the authors are aware that that the physical representation of the wind turbine wake would be improved by a formulation for the mixing length that accounts for local flow gradients. A statement summarizing the intent and limitations of the current approach has been added to the Formulation section. We have also added a derivation of the equations invoking the eddy viscosity hypothesis for both base and wake deficit solution. The conclusions section now includes a statement that points to the development of better mixing length models in future work:

"Future work will consist of comparing the model with RANS, improving the turbulence model without compromising computational cost, implementing a vortex decay model, using the solver for yaw-angle optimizations in a wind plant, and code performance improvements to increase speed."

10. Eq. 13, you can report the explicit formulation ($\Delta((u( = -_{/}\underline{\quad}^{+}{}_{0'+}^{-}$,

This formulation uses the information from the upstream plane and adds a new velocity. This formulation has been improved in the manuscript to denote the current plane (n) and the upstream plan (n-1).

11. L114, provide a reference for the mixing length in the free atmosphere equal to 15 m.

    The references for the formulation (Blackadar 1962 and Sun 2011) were included at the end of the line in the original submission. We have moved the reference next to where the mixing length is defined.

12. Eqs. 14 and 15, I guess rather than $v'$ and $w'$, they should be Δ((($v($ and Δ(($w((.$ Again, more comments on the consequences of replacing the turbine forcing with a velocity perturbation on the momentum and mass budgets might be helpful.

    Thank you for pointing this out, this has been modified in the manuscript.

13. Eqs. 14 and 15 seem different from what reported in Martinez-Tossas 2019, please crosscheck.

    The new equations are written in continuous form. Also, the V and W formulations were switched in the paper from Martinez-Tossas 2019 and corrected in the new one. In the limit of the number of vortices ($N$) going to infinity, the formulations in Martinez-Tossas 2019 should converge to the continuous form in the manuscript.

14. Eq. 17. Cross-check the finite-difference scheme, e.g. there is a second-order approximation of the first derivative, so it should be divided by $2\Delta y$ and $2\Delta z$.

    The reviewer is correct. This was a typo in the manuscript and has been fixed.

15. Eq. 17. Provide the final parabolic equation solved in the code.

    We have re-written the equation in its final form. This is now Equation 20 in the latest version of the manuscript.

16. L 145, forward in time? Maybe forward in the streamwise direction. Please do not mention time to avoid confusion.

    We agree that the term 'time' can be confusing. However, that is the name of the numerical method. We have stated that in the manuscript:

    "This numerical equation is discretized using a ``forward-in-time centered-in-space'' method with the stability criteria shown in Equation 20. We note that the model proposed is steady state and there is no time dependency."

17. L152 – how this grid resolution is obtained? Have you done any study on grid sensitivity?

    Yes, we have included a grid resolution study in the Appendix and have added the following text to the manuscript:

    "Our tests have shown that the implementation has a converged and stable solution when using a grid resolution on the order of $\frac{D}{\Delta y}$10 in the spanwise directions

($y$ and $z$) and $\frac{D}{\Delta x}$ 20 in the streamwise direction. A grid convergence study is shown in Appendix A."

18. Sect. 4.1, How did you set the thrust coefficient of each turbine? Likewise for Sect. 4.2.
The thrust and power coefficients were determined from a lookup table based on the incoming velocity. The discussion on power and thrust coefficients has been expanded and the following sentence was included:

"The power and thrust coefficients are obtained from a lookup table based on the local velocity $\langle \ubavg + \uwavg \rangle_{\rm n-1}$."

The discussion of the effect of yaw angle on power and thrust has also been added to section 2.3.

19. Sect. 4.3, I recommend providing an assessment at the turbine level, considering the data availability from LES.
We have expanded the section and included turbine-specific plots of power for two of the cases.

20. Fig. 8, add the turbine numbers in the color maps. The LES data might be questionable, considering the difference in power for the turbines at the first row. Is there any specific reason, rather than numerical issues? Is there a better LES/RANS dataset to use for this assessment?
We have adjusted the simulations and expanded the discussion. It takes too much space to name the turbines inside the colormap. We have included the ordering of the turbines in the figure legend. One of the nice features of the model presented is that we can adjust the background solution. We now use the time-averaged LES precursor simulation as the background flow solution. This dataset is good because it has a collection of different conditions. The comparison is now much better after using the background flow from the LES.

21. L224, you can say parabolic solution of the streamwise momentum equation of the RANS.
Thanks for the suggestion, we have modified the sentence to:
"The approach uses a hybrid RANS-analytical framework to obtain the wake velocity based on a parabolic solution for the streamwise component of the RANS equations."

---

## Referee Report (RR1)

**Review of *The curled wake model: A three-dimensional and extremely fast steady-state wake solver for wind plant flows*, version R1, by Luis A Martínez-Tossas et al.**

Reviewer: M. Paul van der Laan, DTU Wind Energy

The authors have responded correctly to most of my comments, and they have revised article accordingly. The article can be published as it is but it could be further improved following the remaining comments that I have listed below, all related to the $C$ parameter.

1. The main remaining discussion point where I disagree with the authors is about the usage of the additional constant $C$ in the eddy-viscosity of the mixing length model. It is simply not a free parameter and it can be absorbed in the mixing length parameter to obtain an effect maximum turbulence length scale and von Kármán constant, as shown in the previous review. If the authors prefer to keep using the redundant $C$ parameter, then I strongly suggest that you add a sentence to clarify that the constant $C$ can also be absorbed in the mixing length parameter, but you prefer to use it differently. I am quite sure that experienced readers of your article would have similar thoughts about the $C$ parameter.

   I suspect that the authors have been inspired by a two equation model like the $k$-$\varepsilon$ model, where a $C_\mu$ coefficient/ constant exists in the eddy-viscosity relation ($\nu_T = C_\mu k^2/\varepsilon$). However, $C_\mu$ cannot be absorbed into another existing parameter in the $k$-$\varepsilon$ model and it is therefore an independent coefficient. For example, if one would absord the $C_\mu$ coefficient in the $k$-equation by defining a new $k$: $k_{\text{eff}} = \sqrt{C_\mu} k$. Then it would remain in an $k_{\text{eff}}$-equation as a factor of the source terms ($\sqrt{C_\mu}\,(\mathcal{P} - \varepsilon)$). In addition, a $\sqrt{C_\mu}$ factor would show up in the $\varepsilon$-equation, but this could be replaced by effective $C_{\varepsilon,1}$ and $C_{\varepsilon,2}$ coefficients ($C_{\varepsilon,1,\text{eff}} = C_{\varepsilon,1}\sqrt{C_\mu}$ and $C_{\varepsilon,2,\text{eff}} = C_{\varepsilon,2}\sqrt{C_\mu}$.

2. You have added that the following in Section 2.2: *The constant C is used to account for the additional turbulence introduced by the rotor and the wake.* However, one would expect a lower mixing (or turbulence length scale) in the near wake compared to the atmospheric mixing (at least for neutral conditions), which is exactly what a modified $k$-$\varepsilon$ model as the $k$-$\varepsilon$-$f_P$ model is designed to do, as shown in van der Laan et al. (2015), and further clarified in van der Laan and Andersen (2018) for other modified $k$-$\varepsilon$ models. The value of the maximum length scale of the free atmosphere is quite low (15 m) and would correspond to stable conditions. Note that this maximum length scale can also be interpreted as a proxy for an ABL height, see for example van der Laan et al. (2020). Blackadar (1962) used $\lambda = 0.00027 G/f_c$ to model neutral conditions, which would be nearly twice as large as your chosen value, since $\lambda = 0.00027 \times 10/10^{-4} = 27$ m for typical values of the geostrophic wind speed $G$ and the Coriolis parameter $f_c$. In other words, you could have chosen a higher value of $\lambda$ and then remove $C$ (or set $C = 1$).

**References**

Blackadar, A. K.: The vertical distribution of wind and turbulent exchange in a neutral atmosphere, Journal of Geophysical Research, 67, 3095, 1962.

van der Laan, M. P. and Andersen, S. J.: The turbulence scales of a wind turbine wake: A revisit of extended k-epsilon models, Journal of Physics: Conference Series, 1037, 1, https://doi.org/10.1088/1742-6596/1037/7/072001, 2018.

van der Laan, M. P., Sørensen, N. N., Réthoré, P.-E., Mann, J., Kelly, M. C., Troldborg, N., Schepers, J. G., and Machefaux, E.: An improved $k$-$\varepsilon$ model applied to a wind turbine wake in atmospheric turbulence, Wind Energy, 18, 889, https://doi.org/10.1002/we.1736, 2015.

van der Laan, M. P., Kelly, M., Floors, R., and Peña, A.: Rossby number similarity of an atmospheric RANS model using limited-length-scale turbulence closures extended to unstable stratification, Wind Energy Science, 5, 355–374, https://doi.org/10.5194/wes-5-355-2020, https://wes.copernicus.org/articles/5/355/2020/, 2020.

---

## Editor Decision (ED1)

Wes 2020-86

Overall
- Overall, you have the makings of two good papers here and by shoving it into one paper, you are short-changing some of the depth in analysis on the different components. I would either expand in several areas or potentially cut some content and think about including it in a follow-on paper. For example, the complex terrain – since this is on wake steering, how much value does this add? Couldn't you do a whole additional paper looking at complex terrain and perhaps some application in either design or control? Sections 4.1 and 4.3 seem much more relevant since the main application targeted in this paper is on wake steering
- The model proposed is proposed as an alternative to 1) lower fidelity models (linear engineering models, linear rans (e.g. FUGA)), and 2) higher fidelity models (full steady RANS, etc) and is trying to hit a sweet-spot in terms of the capturing relevant physics at an acceptable computational cost for control and optimization applications. This is clear and articulated at some points in the paper but could be pulled out even more strongly and with more thorough comparison to the state of the art on the former in comparison to both alternatives.
- See more detailed comments by section

Abstract
- some of the results reporting is pretty vague. What does good agreement mean?
- When you say minimizes, that is an exaggeration. There are even simpler models that can do such simulations in fractions of a section
- Saying "about a second on a personal laptop" is vague
- Generally, was the minimization of time an explicit goal in the sense that you optimized aspects of the model parameterization for time minimization, or is it that you were seeking to implement a cost-efficient model that adequately accounts for wake deflection under steering. It seems like the latter is likely the goal and no explicit optimization of the modelling approach is done. It would be better to be explicit that the model is particularly advantageous for addressing specific flow phenomena (such as steering) that are a challenge for conventional engineering flow models…

Introduction
- Okay, I just read the first sentence of the introduction and got more out of that about what the paper is about than the entire abstract. I recommend rewriting the abstract
- Recommend modifying sentence line 18 – qualify minimizing computational cost. You are minimizing cost while doing what? Preserving physics?
- I would like to see a more thorough critique of superposition and where such methods are challenged. It is stated in the beginning that there are differences based on methods but differences aren't necessarily bad, it could be that the differences mean that one model is much better than another… how do they stack up in validation and where in particular are they challenged? Are they challenged more in wake steering compared to normal operating conditions?

- Overall point of paper "The curled wake solver presented in this work focuses on minimizing computational cost and capturing wake steering effects." Should be introduction. Still here, I would not use the term minimize unless you are actively tweaking parameters in a scheme to explicitly minimize the cost while preserving some explicit level of accuracy in wake effects (i.e. the deficit and profile of the wake matches on some statistical factors with agreement of x%)

**Formulation**
- Can you be more explicit in what makes the curl floris standard so much slower? Also, how does it compare to engineering models like floris gaussian or others? Can you somehow quantify the performance of these different models in terms of a two-dimensional perspective on computational cost versus accuracy? The discussion at the end of section 3 seems somewhat incomplete and this is such an important contribution of the overall paper – I would like to see more attention paid to it. I think you are jumping to quickly to the case studies which are arguably less interesting in terms of the core contribution

**Results**
- Again, section 4.1 discussion seems somewhat incomplete. There is a lot of interesting stuff in the results and the plots are great, but there is so little discussion of the results and particularly the results across the different directions – which vary substantially.
- Okay, you are now killing me a little bit… there is so little analysis and discussion relative to the scope of the work being presented in 4.2!

**Conclusions**
- The conclusion is a better summary of the work than the present abstract
- Recommend considering updates to the conclusion following recommended updates in formulation in results section
- Consider discussing a bit more in depth the limitations of the current model and avenues for further work (as its own paragraph)

---

## Author Response (AR2)

**Review of The curled wake model: A three-dimensional and extremely fast steady-state wake solver for wind plant flows, version R1, by Luis A Martínez-Tossas et al.**

Reviewer: M. Paul van der Laan, DTU Wind Energy

The authors have responded correctly to most of my comments, and they have revised article accordingly. The article can be published as it is but it could be further improved following the remaining comments that I have listed below, all related to the C parameter.

1. The main remaining discussion point where I disagree with the authors is about the usage of the additional constant *C* in the eddy-viscosity of the mixing length model. It is simply not a free parameter and it can be absorbed in the mixing length parameter to obtain an effect maximum turbulence length scale and von Kármán constant, as shown in the previous review. If the authors prefer to keep using the redundant *C* parameter, then I strongly suggest that you add a sentence to clarify that the constant *C* can also be absorbed in the mixing length parameter, but you prefer to use it differently. I am quite sure that experienced readers of your article would have similar thoughts about the *C* parameter.

I suspect that the authors have been inspired by a two equation model like the k- $\varepsilon$  model, where a  $C_{\mu}$  coefficient/constant exists in the eddy-viscosity relation ( $v_T = C_{\mu}k^2/\varepsilon$ ). However,  $C_{\mu}$  cannot be absorbed into another existing parameter in the k- $\varepsilon$  model and it is therefore an independent coefficient. For example, if one would absorb the  $C_{\mu}$  coefficient in the k-equation by defining a new k: keff =  $\mu C k$ . Then it would remain in an  $k_{eff}$  equation as a factor of the source terms ( $C_{\mu} (P - \varepsilon)$ ). In addition,  $a_{\mu} C$  factor would show up in the  $\varepsilon$ -equation, but this could be replaced by effective  $C_{\varepsilon,1}$  and  $C_{\varepsilon,2,eff} = C_{\varepsilon,2} C_{\mu}$ .

We agree with the reviewer with the fact that there are many ways of implementing, developing and interpreting a turbulence model in RANS. We also agree with the fact that the constant C could also be absorbed in the mixing length in the current formulation. The inspiration behind the model comes from the idea of expressing the eddy-viscosity as a function of the background flow eddy- viscosity. The purpose of the constant C is to scale the turbulent viscosity such that it takes into account the wake added turbulence. The constant C modifies the total turbulent viscosity and not just the mixing length. For example, if we had a different eddy viscosity model, the formulations for the constant would be the same as presented in this work. Appendix B: Turbulence modeling, shows a derivation including C as a function of space in the formulations. Equation B6 shows the functional form of C. We make the approximation of choosing C as a constant to avoid the computational cost of solving another set of equations. Further work should focus on improving the turbulence model in this work.

2. You have added that the following in Section 2.2: *The constant C is used to account for the additional turbulence introduced by the rotor and the wake.* However, one would expect a lower mixing (or turbulence length scale) in the near wake compared to the atmospheric mixing (at least for neutral conditions), which is exactly what a modified *k*- $\varepsilon$  model as the *k*- $\varepsilon$ -*f*P model is designed to do, as shown in van der Laan et al. (2015), and further clarified in van der Laan and Andersen (2018) for other modified *k*- $\varepsilon$  models. The value of the maximum length scale of the free atmosphere is quite low (15 m) and would correspond to stable conditions. Note that this maximum length scale can also be interpreted as a proxy for an ABL height, see for example van der Laan et al. (2020). Blackadar (1962) used  $\lambda = 0.00027 \, G/f_c$  to model neutral conditions, which would be nearly twice as large as your chosen value, since  $\lambda = 0.00027 \, \times 10/10^{-4} = 27 \, m$  for typical values of the geostrophic wind speed *G* and the Coriolis parameter *f*c. In other words, you could have chosen a higher value of  $\lambda$  and then remove *C* (or set *C* = 1).

**The curled wake model: A three-dimensional and extremely fast steady-state wake solver for wind plant flows**

Luis A Martínez-Tossas1, Jennifer King1, Eliot Quon1, Christopher J Bay1, Rafael Mudafort1, Nicholas Hamilton1, Michael F Howland2, and Paul A Fleming1

1National Renewable Energy Laboratory, Golden, CO USA

[revised manuscript text omitted]

---

## Author Response (AR3)

Wes 2020-86
**Overall**

Overall, you have the makings of two good papers here and by shoving it into one paper, you are short-changing some of the depth in analysis on the different components. I would either expand in several areas or potentially cut some content and think about including it in a follow-on paper. For example, the complex terrain - since this is on wake steering, how much value does this add? Couldn't you do a whole additional paper looking at complex terrain and perhaps some application in either design or control? Sections 4.1 and 4.3 seem much more relevant since the main application targeted in this paper is on wake steering

The model proposed is proposed as an alternative to 1) lower fidelity models (linear engineering models, linear rans (e.g. FUGA)), and 2) higher fidelity models (full steady RANS, etc) and is trying to hit a sweet-spot in terms of the capturing relevant physics at an acceptable computational cost for control and optimization applications. This is clear and articulated at some points in the paper but could be pulled out even more strongly and with more thorough comparison to the state of the art on the former in comparison to both alternatives.
See more detailed comments by section

We thank the editor for their positive feedback and suggestions. We have revised the abstract to better describe the work in the paper. We want to keep the article and its content in the same publication. This article shows the mathematical method in detail with rigorous derivations that are new and have not been done before. We also believe that the test cases are an excellent way of showing the extent of the model capabilities. We would like to pursue future work in complex terrain and either design or control, however, this is out of the scope of the current work within our organization. This paper has also undergone an extensive review process with 2 revisions as part of the WES journal review and 3 revisions within our communications department. The new version of the manuscript meets all our internal requirements, and the external reviewer feedback has been addressed in detail.

The abstract of the text has been rewritten. The new abstract is:

"Wind turbine wake models typically require approximations, such as wake superposition and deflection models, to accurately describe wake physics. However, capturing the phenomena of interest, such as the curled wake and interaction of multiple wakes, in wind power plant flows comes with an increased computational cost. To address this, we propose a new hybrid method that uses analytical solutions with an approximate form of the Reynolds-averaged Navier-Stokes equations to solve the time-averaged flow over a wind plant. We compare results from the solver to supervisory control and data acquisition data from the Lillgrund wind plant obtaining wake model predictions which are generally within one standard deviation of the mean power data. We perform simulations of flow over the Columbia River Gorge to demonstrate the capabilities of the model in complex terrain. We also apply the solver to a case with wake steering, which agreed well with large-eddy simulations. This new solver reduces the time–and therefore the related cost–it takes to simulate a steady-state wind plant flow (on the order of seconds using one core). Because the model is

computationally efficient, it can also be used for different applications including wake steering for wind power plants and layout optimization."

**Abstract**
some of the results reporting is pretty vague. What does good agreement mean?

In most cases, we defined good agreement for power predictions as being within one standard deviation from the SCADA or high-fidelity simulations. We have eliminated some statements in the paper where "good agreement" was vague.

When you say minimizes, that is an exaggeration. There are even simpler models that can do such simulations in fractions of a section

To our knowledge, there is no other model that can solve a similar set of equations with a reduced computational cost. We have shown in section 3.1 that the problem of solving a simplified form of the RANS equations is reduced to a computational cost of order N, where N is the number of grid points in the domain. We acknowledge, that there is always room for improvement and to reduce the computational cost. We have replaced the word 'minimize' with 'reduce':

"This new solver reduces the time—and therefore the related cost—it takes to simulate a steady-state wind plant flow (on the order of seconds using one core)."

Saying "about a second on a personal laptop" is vague

We cannot use an exact number to describe computational time. "Order of seconds" is a good reference to use. The exact time varies depending on the computer, processor architecture, clock speed, software version, etc. To clarify the statement, we have rephrased it:

"on the order of seconds using one core"

Generally, was the minimization of time an explicit goal in the sense that you optimized aspects of the model parameterization for time minimization, or is it that you were seeking to implement a cost-efficient model that adequately accounts for wake deflection under steering. It seems like the latter is likely the goal and no explicit optimization of the modelling approach is done. It would be better to be explicit that the model is particularly advantageous for addressing specific flow phenomena (such as steering) that are a challenge for conventional engineering flow models...

The model was developed with the idea of saving computational cost. Many of the derivations and approximations were done with the explicit goal of reducing the computational cost. We originally developed the model to address wake steering. However, after different tests as shown in the paper, we have learned that the model can also be used for different physics phenomena, including shear, veer, complex terrain, etc.

**Introduction**

Okay, I just read the first sentence of the introduction and got more out of that about what the paper is about than the entire abstract. I recommend rewriting the abstract

Yes, we have re-written the abstract following the recommendations.

Recommend modifying sentence line 18 - qualify minimizing computational cost. You are minimizing cost while doing what? Preserving physics?

Yes, the model intends to preserve physics by solving the hybrid analytical-RANS set of equations. We have added the following statement to clarify that:

"This solver uses a hybrid RANS-analytical framework that aims to minimize computational cost while still preserving physics from the RANS equations."

I would like to see a more thorough critique of superposition and where such methods are challenged. It is stated in the beginning that there are differences based on methods but differences aren't necessarily bad, it could be that the differences mean that one model is much better than another... how do they stack up in validation and where in particular are they challenged? Are they challenged more in wake steering compared to normal operating conditions?

That is a good point, and we have looked into wake superposition models in other work. We have learned that wake superposition models can have a big impact on the simulated power of a wind plant with relative errors that go up to 100% per turbine. Errors are typically higher when the number of wakes being superposed is higher, such as in the fully developed region of a wind plant. We have included a new reference in the discussion:

"Hamilton, N., Bay, C. J., Fleming, P., King, J., and Martínez-Tossas, L. A.: Comparison of modular analytical wake models to the Lillgrund wind plant, Journal of Renewable and Sustainable Energy, 12, 053 311, https://doi.org/10.1063/5.0018695, 2020."

Overall point of paper "The curled wake solver presented in this work focuses on minimizing computational cost and capturing wake steering effects." Should be introduction. Still here, I would not use the term minimize unless you are actively tweaking parameters in a scheme to explicitly minimize the cost while preserving some explicit level of accuracy in wake effects (i.e. the deficit and profile of the wake matches on some statistical factors with agreement of %)

We have replaced the word 'minimize' with 'reduce'.

**Formulation**

Can you be more explicit in what makes the curl floris standard so much slower? Also, how does it compare to engineering models like floris gaussian or others? Can you somehow quantify the performance of these different models in terms of a twodimensional perspective on computational cost versus accuracy? The discussion at the end of section 3 seems somewhat incomplete and this is such an important contribution of the overall paper-I would like to see more attention paid to it. I think you are jumping to quickly to the case studies which are arguably less interesting in terms of the core contribution

The reason for the significant speedup has to do with the current implementation of the model. In the standard FLORIS implementation, we need to compute every wake individually in the domain. That means that the equation is solved as many times as there are turbine in the domain. After the calculations of individual wakes, they need to be superposed, which can also be computationally expensive. We have included the following text in the manuscript:

"Significant speedup is expected in the presently proposed curled wake model formulation compared to the standard FLORIS implementation. The standard FLORIS implementation solves Equation 16 for every turbine in the domain individually and then superposes the solutions. This superposition approach results in an increased computational cost, especially when more turbines are included, as well as wake superposition uncertainty.

**Results**

· Again, section 4.1 discussion seems somewhat incomplete. There is a lot of interesting stuff in the results and the plots are great, but there is so little discussion of the results and particularly the results across the different directions - which vary substantially.

We agree that this work has opened the door for more comparisons and future work. We have opted for a concise discussion of the results in this section. The goal of this paper was to present the derivations in detail and demonstrate a few cases where the model can be used. Future work should focus on more in-depth sensitivity studies and other comparisons.

- Okay, you are now killing me a little bit... there is so little analysis and discussion relative to the scope of the work being presented in 4.2!

This section is meant to be a test case to demonstrate the capability of the solver in dealing with complex terrain. We are interested in pursuing research in this area, but unfortunately, at the moment, this work is out of the scope of current projects within our organization.

Conclusions
The conclusion is a better summary of the work than the present abstract
Recommend considering updates to the conclusion following recommended updates in formulation in results section
Consider discussing a bit more in depth the limitations of the current model and avenues for further work (as its own paragraph)

We have revised the abstract following the recommendations to better summarize the work in the article. We have also updated the conclusions to include the limitations of the model. A new paragraph has been added to the conclusion:

"Some of the limitations from the different approximations of the model include: a turbulence model mixing length that only depends on the vertical coordinate, a linearized solution of the vortices from curl that do not decay, a near wake formulation is missing, and there is no pressure term in the equations. These approximations were done in order to reduce the computational cost. Future work will focus on comparing the model with RANS, improving the turbulence model without compromising computational cost, improving the near wake, implementing a vortex decay model, using the solver for yaw-angle optimizations in a wind plant, and improving code performance to increase speed."

---

## Author Response (AR4)

Wes 2020-86

Comments to the Author:

The editors comments regarding section 4.2 still need to be sufficiently addressed. The current section 4.2 does not have adequate substantive content to be included - figures on their own are not sufficient for validation, demonstration of novel scientific findings, etc. The authors are encouraged to either provide:

- a minimum quantitative analysis of errors in section 4.2 (apparently they have the LES, since they took the velocity from it??)

- or remove the section

- or shorten, retain only one illustrative figure showing complex terrain and integrate in other section (not as part of results)

We have shortened section 4.2 and included it as part of Section 3. The new section presents only a brief overview of the complex terrain capability and only one figure is used to show the capability.